# SPEECH-DRAME: A FRAMEWORK FOR HUMAN-ALIGNED BENCHMARKS IN SPEECH ROLE-PLAY

## ABSTRACT

Role-play has become a key testbed for generative models, expanding from text-only dialogue to multimodal interaction. Extending role-play to speech captures prosody, emotion, and delivery, but also poses new evaluation challenges. Current pipelines often use audio large language models (ALLMs) as zero-shot judges, which miss paralinguistic cues, collapse multiple aspects into coarse scores, and rely on synthetic speech references that fail to reflect real-world roles. We present **Speech-DRAME**, a unified framework that contributes at three levels: (i) *Speech-DRAME-EvalBench*, an evaluation benchmark with bilingual human-annotated data and protocols for training and testing speech evaluation models (SEMs), (ii) `DRAME-Eval`, a fine-tuned evaluation model, which substantially outperforms zero-shot and few-shot ALLMs, and (iii) *Speech-DRAME-RoleBench*, a speech role-play benchmark that leverages `DRAME-Eval` as an automatic judge to compare speech foundation models (SFMs). Speech-DRAME distinguishes between two complementary evaluation strategies: *Archetype Evaluation*, a top-down approach measuring adherence to broad role archetypes, and *Realism Evaluation*, a bottom-up approach grounded in real human speech that emphasizes nuanced role quality. Compared to zero-shot ALLM judges, `DRAME-Eval` achieves stronger agreement with human ratings (Pearson correlation from 0.480 to 0.629 in archetypes, and 0.390 to 0.625 in realism). By integrating transparent benchmark resources, modeling approaches, and system-level evaluation, Speech-DRAME provides the first comprehensive, reproducible foundation for assessing spoken role-play.

## 1 INTRODUCTION

Role-play has been recently recognized a cornerstone of interactive systems, from initial text-based focuses to the growing ecosystem of multimodal dialogue agents (Chen et al., 2024a; Tseng et al., 2024; Dai et al., 2025; Ma et al., 2024; Jiang et al., 2025). With the rapid advancement of large language models (LLMs), role-play has evolved beyond scripted interactions into dynamic character-driven conversations, supporting applications in education, entertainment, and human–AI collaboration (Wu et al., 2024; Agatsuma et al., 2024; Fung & Laing, 2024). Extending role-play to speech-based interaction offers an even richer communication channel, capturing prosody, emotion, and character consistency in ways that text alone cannot (Ding et al., 2025; Huang et al., 2025a).

Evaluating speech role-play, however, remains a major challenge. Current pipelines often rely on ALLMs as judges, typically in a zero-shot setting (Huang et al., 2025b; Chiang et al., 2025; Jiang et al., 2025; Zhang et al., 2025a). While this strategy can provide rough ratings, it falls short for speech role-play:

- it lacks sensitivity to paralinguistic cues such as intonation, pacing, and emotional dynamics that define role delivery (Zhang et al., 2025a; Huang et al., 2025b; Chiang et al., 2025; Zhou et al., 2025a),
- it collapses diverse aspects of performance into limited coarse dimensions, obscuring critical dimensions of role-play quality (Chiang et al., 2025; Huang et al., 2025b), and
- it depends on synthetic speech references rather than real human speech (Zhang et al., 2025a; Jiang et al., 2025), limiting its ability to reflect real-world diversity in speech role-play.

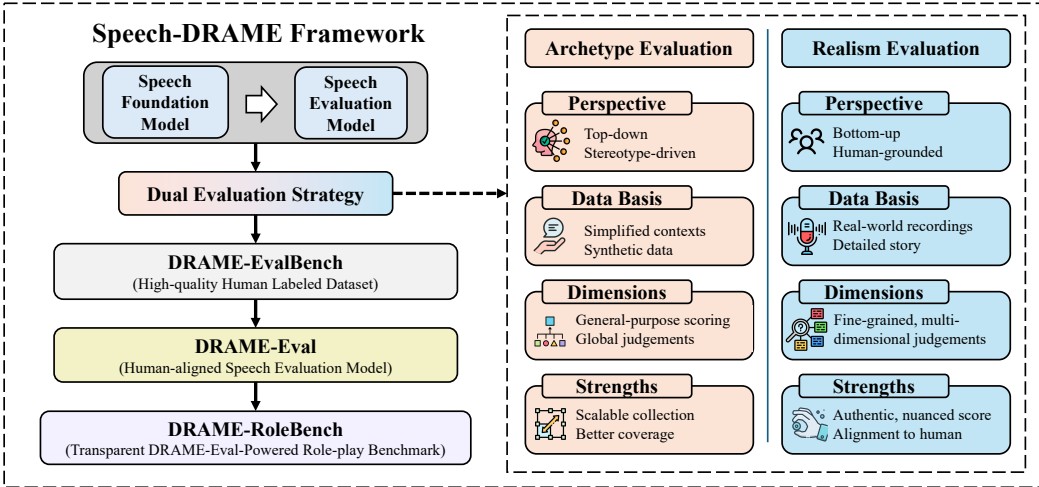

Figure 1: Speech-DRAME formalizes speech role-play with a dual evaluation strategy: **Archetype Evaluation** (top-down, stereotype-driven) and **Realism Evaluation** (bottom-up, human-grounded). Built on these, **DRAME-EvalBench** provides human-annotated data, **DRAME-Eval** aligns SEMs to perception, and **DRAME-RoleBench** benchmarks SFMs with automatic judges.

To overcome these limitations, we present **Speech-DRAME** (Speech Detailed Role-play Assessment with Modeling and Evaluation), a unified framework that moves beyond synthetic or judge-only pipelines. Speech-DRAME is distinguished by three aspects: [i] a dual evaluation paradigm that balances archetype-driven generality with human-grounded realism (see Sec. 3.2), [ii] human annotations that provide perceptual grounding absent in prior resources (see Sec. 4), and [iii] evaluation-model alignment that enables not only benchmarking of speech foundation models (SFMs) but also systematic analysis of the evaluation models (SEMs) themselves (see Sec. 5). These aspects establish Speech-DRAME as a comprehensive and trustworthy framework for measuring and improving speech role-play.

As illustrated in Fig. 1, Speech-DRAME contributes at three levels: (i) **DRAME-EvalBench**, an evaluation benchmark consisting of human-annotated datasets for archetype and realism role-play; (ii) **DRAME-Eval**, a fine-tuned SEM trained on EvalBench that surpasses zero-shot and few-shot ALLMs; and (iii) **DRAME-RoleBench**, a speech role-play benchmark that leverages DRAME-Eval as an automatic judge to systematically compare SFMs. By combining dual evaluation strategies, human-anchored supervision, and evaluation-model alignment, Speech-DRAME delivers systematic insights into both role-play systems and evaluation models themselves. Our contributions are four-fold:

- We introduce a **formal task definition** of speech role-play and its evaluation via speech foundation models (SFMs) for generation and speech evaluation models (SEMs) for assessment, unifying linguistic and paralinguistic dimensions.

- We construct **DRAME-EvalBench**, the first evaluation benchmark with human annotations under both archetype- and realism-based strategies, supporting the training and testing of SEMs.

- We propose **DRAME-Eval**, a fine-tuned SEM trained on EvalBench that substantially outperforms zero-shot and few-shot ALLMs, demonstrating the importance of human supervision for evaluation models.

- We establish **DRAME-RoleBench**, a speech role-play benchmark that leverages DRAME-Eval as an automatic judge to systematically compare both end-to-end and cascaded SFMs across multiple role quality dimensions.[1]

Through this work, we aim to move from acting in text to acting in speech, equipping the community with the models, evaluation strategies, and benchmarks needed to measure and improve speech role-play.

---

[1] Artifacts including datasets, evaluation recipes, trained models, and benchmark splits will be open-sourced after the review period.

## 2 RELATED WORKS

**Text-based role-play evaluation.** Early research focused on text-only dialogue benchmarks. Datasets such as RoleLLM, CharacterBox, and RoleMRC (Wang et al., 2024; Tu et al., 2024; Wang et al., 2025a; Shen et al., 2023; Lu et al., 2025) test whether models can sustain persona consistency and character fidelity, using LLM-as-judge pipelines for scalability. However, studies like PersonaEval (Zhou et al., 2025a) have shown that LLM-as-judge approaches are far from perfect, struggling with tasks as basic as tracking speaker identity. This highlights broader concerns about reliability even in text settings, which become only more severe when extended to speech, where prosody, emotion, and vocal style play a central role.

**Speech-native benchmarks.** More recently, a number of resources have been introduced to evaluate spoken role-play agents. OmniCharacter-10K (Zhang et al., 2025a) emphasizes character voice traits, while others (Liu et al., 2025; Hou et al., 2025; Du et al., 2025; Chen et al., 2024b) extend evaluation toward vocal communication, assistant capability, and multi-turn realism. Omni models such as StepAudio and KimiAudio (Huang et al., 2025a; Ding et al., 2025) also include role-play scenarios in their evaluation. SpeechRole and VStyle (Jiang et al., 2025; Zhan et al., 2025) are closer to our work, offering datasets that test interaction ability, expressiveness, and fidelity. However, these resources are primarily built from LLM-generated dialogues and TTS outputs, with limited human validation and a heavy reliance on proprietary ALLM-as-judge pipelines. As a result, they lack grounding in diverse human speech and typically apply only coarse rubrics.

**Automatic judging with ALLMs.** In parallel, advances in automatic judging highlight both opportunities and limitations. Audio-capable LLMs such as AudioJudge and Audio-Aware LLMs as Judges (Manakul et al., 2025; Chiang et al., 2025) can assess style, emotion, and delivery, while systems like Auto-ATT (Wang et al., 2025c) scale this to Audio Turing Tests. Yet studies consistently reveal high prompt sensitivity and shallow capture of paralinguistic cues, suggesting that zero-shot judging remains insufficient for nuanced role-play evaluation.

**Expressive TTS evaluation.** Recent work has explored evaluation of instruction-following and style-controlled TTS. InstructTTSEval (Huang et al., 2025b) explicitly includes role-play scenarios, while EmergentTTS-Eval (Manku et al., 2025) and discrete speech LM benchmarks (Wang & Székely, 2024) emphasize prosody, spontaneity, and speaker consistency. These efforts push beyond intelligibility and naturalness, but still center on synthetic outputs rather than real human role-play.

**Our distinction.** In contrast, Speech-DRAME unifies these considerations through three elements: (i) a dual evaluation paradigm that combines archetype-driven generality with realism grounded in authentic human speech, (ii) human annotations that provide a reliable perceptual gold standard, and (iii) alignment of SEMs to human ratings, enabling systematic evaluation of both SFMs and the evaluation models themselves. This combination moves beyond synthetic benchmarks or zero-shot judging to provide a comprehensive and trustworthy framework for speech role-play assessment.

## 3 SPEECH ROLE-PLAY: FORMULATION AND EVALUATION STRATEGIES

### 3.1 PROBLEM FORMULATION

**Speech role-play.** We formalize *speech role-play* as a conditional speech generation problem grounded in a character profile and scene description. Given a dialogue context $C_{\text{speech}}$, a character profile $C_{\text{profile}}$, and a scene specification $C_{\text{scene}}$, a *speech foundation model* (SFM) generates a role-played speech response $S_r$:

$$S_r = \text{SFM}(C_{\text{speech}}, C_{\text{profile}}, C_{\text{scene}}). \quad (1)$$

Here $S_r$ represents the spoken output (waveform or latent representation). This definition captures both the linguistic dimension (coherence with context and scene) and the paralinguistic dimension (prosody, style, emotion) expected of the role.

**Evaluation.** In principle, a *speech evaluation model* (SEM) can assess a generated response using the same conditioning signals:

$$E_r = \text{SEM}(S_r, C_{\text{speech}}, C_{\text{profile}}, C_{\text{scene}}). \quad (2)$$

However, in this work we focus on the *single-turn setting*, where evaluation is defined at the response level and does not explicitly depend on the preceding dialogue context:

$$E_r = \text{SEM}(S_r, C_{\text{profile}}, C_{\text{scene}}). \tag{3}$$

This formulation is intentionally modular: a dialogue-level evaluation can be obtained by aggregating single-turn scores across turns (e.g., averaging, recency-weighted pooling). We leave the detailed discussion of this multi-turn extension to Appendix C.

## 3.2 ROLE-PLAY EVALUATION WITH TWO STRATEGIES

Speech role-play evaluation can be framed in multiple ways depending on the available data and the desired granularity of analysis. Within Speech-DRAME, we formalize two distinct strategies inspired by sociology theories[2], including a top-down, archetype-oriented option using synthetic speech and a bottom-up, realism-oriented option from real human speech. Rather than merging these into a single rubric, we treat them as separate perspectives, each with unique advantages and limitations. Figure 1 summarizes the key contrasts, while further implementation details are provided in the Appendix D- G.

**Archetype-based Role-play Evaluation** (examples in Appendix D.2). The first strategy follows a *top-down* design, inspired by prior text-based role-play benchmarks. Here, role-play is judged with respect to *stereotypical archetypes* (e.g., "firefighter" or "ER doctor"), using scene contexts (e.g., "comfort a child trapped in fire" or "ask for help in an serious operation"). This approach provides accessible, general-purpose scoring that is scalable to a wide range of scenarios, making it suitable for large-scale benchmarking. However, its reliance on broad stereotypes and global impressions limits its ability to capture fine-grained aspects of delivery (e.g., subtle prosodic shifts, nuanced emotional control) (Holt & Lotto, 2010). In practice, this strategy mirrors simplified annotation settings and is particularly useful when annotation resources are limited (Gemmeke et al., 2017).

**Realism-based Role-play Evaluation** (examples in Appendix F.5). The second strategy adopts a *bottom-up* design, motivated by real-world performance and real human speech. Instead of relying solely on stereotypes, this approach grounds evaluation in recordings from professional and non-professional speakers, drawn from realistic dialogue and media sources. Judgments in this setting are more fine-grained and multi-dimensional, addressing aspects such as prosodic dynamics, emotional fidelity, character consistency, and contextual fit. While this strategy requires more human annotation and careful curation, it provides benchmarks that reflect how role-play is perceived in practice. It thus offers a richer framework for analyzing model outputs in settings where realism and subtlety are essential.

## 4 DRAME-EVALBENCH DATA CURATION

This section details the **DRAME-EvalBench** data curation process. Our data curation methodology follows a two–stage design: (i) *task preparation*, which specifies the roles, scenes, and evaluation settings used to elicit speech; and (ii) *human annotation*, which provides reliable ground-truth judgments on generated or real human speech.

**Task Preparation.** We employ complementary strategies: a *top–down* archetype-based approach, rooted in stereotypes and generalized role expectations, and a *bottom–up* realism-based approach, derived from real human speech. Both follow a modular pipeline of role/scene definition, prompt expansion, and speech rendering or retrieval, ensuring scalability and validity.

**Human Annotation.** All collected audio undergoes a standardized annotation protocol: (1) an initial semantically-based screening to exclude invalid clips; (2) rubric-based scoring across multiple perceptual dimensions; and (3) meta-annotations such as confidence or prompt-quality judgments. Full rubrics are provided in the appendices E & G.[3]

Table 1 presents a summary of the curated **DRAME-EvalBench** dataset, where we further elaborate the details of archetype and realism evaluation data collection in following subsections, respectively.

---

[2]Please refer to Appendix B for detailed discussion of the dual evaluation design.

[3]All data collection follows in high standard ethical consideration as outlined in Appendix I.

Table 1: Data summary of **DRAME-EvalBench** for archetype and realism evaluation. We provide a comparison of sources, evaluation dimensions (annotated speech role-play dimensions), supported languages, total samples, split details and scenarios for benchmarking.

|  | **Archetype Evaluation** | **Realism Evaluation** |
|---|---|---|
| **Sources** | SFMs | Media, open-source data, new recordings |
| **Eval. Dimensions** | General (3) | Detailed (10) |
| **Languages** | Mandarin & English | Mandarin & English |
| **Total Samples** | 8,280 | 15,000 |
| **Splits** | 6,780 train
1,500 test | 12,000 train
2,000 base test
1,000 real-recording test |

For benchmark purpose, we provide an official train-test split for the human annotated dataset for **DRAME-EvalBench** so that following researchers can use the dataset to train and evaluate their SEMs for speech role-play.

## 4.1 ARCHETYPE DATA COLLECTION

Archetype evaluation adopts a top–down perspective from sociology and role theory (Goffman, 1949; Campbell, 2008), abstracting role-play into broad and recognizable categories.

**Role and Scene Dimensions.** Roles are operationalized through seven categories: (1) functional occupations in daily life, (2) functional occupations in fictional or fantasy settings, (3) social identities, (4) named or specific characters, (5) personality traits, (6) relative relationship, (7) basic events. Scenes are anchored by everyday events or state changes that evoke stereotypical reactions. Details of template design and alternative prompting strategies are given in Appendix D.[4]

**Data Sources.** The data is collected from a range of speech foundation models, including both cascaded and end-to-end models. We limit that only audio prompts are used as the input but adopt different prompt strategies for best quality with regards to different models. Details about the selected models and prompting details are discussed in Appendix D.4.

**Annotation Protocol and Evaluation Dimensions.** Annotation follows two stages: an initial semantic check (i.e., *Content Pass*) to filter out instruction non-compliance or audio defects, followed by rubric-based scoring on three dimensions: *Audio Quality*, *Human-likeness*, and *Appropriateness*. Confidence judgments are also recorded. Detailed rubric scales are in Appendix E.

**Archetype Evaluation Data Summary.** The resulting archetype evaluation dataset comprises 552 distinct scenarios for both Mandarin and English, evaluated across eight models,[5] yielding 8,280 samples. In total, the dataset are split into 6,780 for training and 1,500 for testing. For benchmarking, an additional 1,250 scenarios spanning seven role/scene categories are prepared to facilitate archetype role-play benchmarking. More detailed statistics and analyses are provided in Appendix J.

## 4.2 REALISM DATA COLLECTION

Realism evaluation adopts a bottom–up perspective, emphasizing authentic delivery, prosody, and contextual fit.

**Data Sources and Negative Samples.** We curate speech from (i) ground-truth media segments, (ii) retrieved character profiles, and (iii) generated local scene descriptions. To stress-test evaluation models, we construct contrastive mismatches: (i) negative local scenes with varying degrees of mismatch, and (ii) negative character profiles created by re-synthesizing transcripts with TTS systems. Appendix F details the prompt design and TTS generation details.

**Annotation Protocol and Evaluation Dimensions.** Annotators judge whether performances align with the *Character Profile* and *Local Scene*, using rubrics for *Prosodic Dynamics (i.e., Pitch Variation, Rhythmic Naturalness, Stress & Emphasis), Emotional Expressiveness (with progressive gating*

---

[4]We note that some archetypes (e.g., firefighter) inevitably reflect cultural stereotypes. These are used solely for evaluation scalability and do not reflect normative judgments.

[5]The model list is discussed in Appdenix D.4. Note we only include seven models for English due to the deactivation of ChatGPT-4O advanced voice model.

*over three sub-dimensions: Emotion Accuracy, Emotion Intensity, Emotion Transition)*, *Character Consistency (i.e., Voice Identity Matching and Traits Embodiment)*, *Contextual Relevance (i.e., Local Scene Fit and Global Story Coherence)*, and *Semantic Match*. A prompt quality check and confidence ratings are also collected. Full scales and gating details are in Appendix G.

**Real-world Recoding.** As noted in Sec. 2, most prior efforts have relied heavily on synthetic data, mostly from TTS systems based on curated professional role-play voices. While such data can achieve high fidelity, it often lacks the diversity and spontaneity of real-world delivery, limiting the generalizability of evaluation models. To address this gap, we expand realism evaluation beyond synthetic and curated voices by incorporating newly recorded speech from both professional and amateur voice actors. This design introduces a broader spectrum of delivery styles, from polished professional acting to more natural and less controlled amateur performance.

**Realism Evaluation Data Summary.** The resulting realism evaluation dataset is constructed from a combination of curated media sources, including internal collections, public datasets (e.g., NC-SSD (Liu et al., 2024b)), and original recordings collected for this project.[6] We compile 15k annotated role-play samples: 9k in Mandarin and 5k in English. The dataset is divided into 12k samples for training, 2k mixed samples for the base test set, and 1k real-recording samples for the test set. The first test set supports general evaluation using our proposed pipeline with negative samples, while the second focuses exclusively on human speech performed by both professional and amateur voice actors. For benchmarking, 2,000 scenarios from the realism dataset are designated for realism role-play benchmarking. Detailed statistics and analyses are provided in Appendix K.

## 5 DRAME-EVALBENCH (ROLE-PLAY EVALUATION MODELING)

### 5.1 EXPERIMENTAL SETUP

Building on the two proposed datasets, we conduct experiments under three settings: zero-shot, few-shot, and finetuning, using pre-trained ALLMs. We formulate the task as a multi-metric estimation problem (Tjandra et al., 2025; Yao et al., 2025; Shi et al., 2025b;a), where a single model is trained or prompted to predict multiple evaluation dimensions.

**Zero-shot and Few-shot ALLMs.** We first evaluate the zero-shot capabilities of both proprietary and public models, including `Gemini2.5Pro`, `GPT4o-audio`, `KimiAudio`, `Qwen2Audio`, and `Qwen2.5Omni`. All evaluations follow a unified prompting template that provides audio input alongside scene and character context.

For proprietary models, we report the average of five independent runs to reduce variance and obtain more stable predictions. For open-source models, we observed weaker instruction-following ability and limited adaptation to unfamiliar tasks. To mitigate this, we did not request simultaneous prediction of multiple metrics. Instead, we queried one metric at a time. Following Song & Bahri (2025); Lukasik et al. (2024), we interpret model outputs through token distribution over discrete labels (`[1]`-`[5]`) and compute the expectation as the final score.

For few-shot experiments, we focus primarily on proprietary ALLMs, which demonstrate stronger instruction-following capabilities and have been widely used as judges in prior work. For each test query, we provide three additional annotated samples from the training set as in-context exemplars.[7]

**Finetuning Model: `DRAME-Eval`.** In addition to zero-shot and few-shot evaluations, we fine-tune a dedicated evaluation model, denoted as `DRAME-Eval` for archetype and realism role-play evaluation, respectively. We initialize from `Qwen2Audio-7B-Instruct` (Chu et al., 2024), which provides a strong foundation for multi-modal audio-text reasoning. To better leverage the multiple human annotations available in our datasets, we adopt a sampling-based training strategy: at each step, one annotator label is randomly selected, following practices in label distribution learning (Geng, 2016; Wen et al., 2023). This stochastic supervision implicitly exposes the model to the empirical annotation distribution, while during inference we output the full distribution over discrete labels (e.g., `[1]`-`[5]`) and compute the expectation as the final score, consistent with recent

---

[6]All data were obtained under appropriate licenses and are legally available for research release.

[7]Please find our detailed settings such as prompts, decoding algorithms in Appendix L.1 with additional ablation analysis in Appendix L.2.

Table 2: Archetype role-play evaluator performance. We report classification accuracy for *Content Pass* and Pearson correlation coefficients for other dimensions. The Avg. column is the average of all correlation coefficients.

| Config. | Model | Content Pass Acc. | Audio Quality | Human Likeness | Appropriateness | Avg. |
|---------|-------|-------------------|---------------|----------------|-----------------|------|
| Zeroshot | Gemini2.5Pro | 89.3% | 0.465 | 0.483 | 0.492 | 0.480 |
| | GPT4o-audio | 89.9% | 0.395 | 0.483 | 0.422 | 0.433 |
| | KimiAudio | 71.7% | 0.089 | 0.132 | 0.121 | 0.114 |
| | Qwen2Audio | 86.7% | 0.078 | 0.210 | 0.200 | 0.163 |
| | Qwen2.5Omni | 84.1% | -0.023 | 0.039 | 0.042 | 0.019 |
| Fewshot | Gemini2.5Pro | 91.1% | 0.477 | 0.492 | 0.517 | 0.495 |
| | GPT4o-audio | 89.9% | 0.296 | 0.374 | 0.377 | 0.349 |
| Finetune | DRAME-Eval | **93.6%** | **0.682** | **0.680** | **0.525** | **0.629** |

Table 3: Realism role-play evaluator performance. We report Pearson correlation coefficients for all dimensions. The Avg. column is the average of all dimensions.

| Config. | Model | Prosodic Dynamics | | | Emotional Expressiveness | | | Character Consistency | | Contextual Relevance | | Avg. |
|---------|-------|-------|--------|--------|----------|-----------|------------|----------|--------|-------|--------|------|
| | | Pitch | Rynthm | Stress | Accuracy | Intensity | Transition | Identity | Traits | Local | Global | |
| Zeroshot | Gemini2.5Pro | 0.319 | 0.352 | 0.336 | 0.391 | 0.316 | 0.336 | 0.529 | 0.509 | 0.349 | 0.465 | 0.390 |
| | GPT4o-audio | 0.132 | 0.210 | 0.179 | 0.323 | 0.258 | 0.273 | 0.360 | 0.397 | 0.277 | 0.440 | 0.284 |
| | KimiAudio | 0.045 | 0.033 | 0.065 | 0.018 | 0.020 | 0.024 | 0.072 | 0.013 | 0.009 | 0.026 | 0.032 |
| | Qwen2Audio | 0.029 | 0.030 | 0.051 | -0.006 | -0.007 | -0.002 | 0.009 | -0.023 | 0.056 | -0.003 | 0.013 |
| | Qwen2.5Omni | 0.124 | 0.073 | 0.102 | 0.227 | 0.208 | 0.226 | 0.151 | 0.291 | 0.143 | 0.203 | 0.175 |
| FewShot | Gemini2.5Pro | 0.380 | 0.415 | 0.406 | 0.418 | 0.354 | 0.354 | 0.596 | 0.524 | 0.376 | 0.497 | 0.432 |
| | GPT4o-audio | 0.236 | 0.264 | 0.230 | 0.417 | 0.344 | 0.412 | 0.422 | 0.514 | 0.332 | 0.498 | 0.367 |
| Finetune | DRAME-Eval | **0.621** | **0.627** | **0.631** | **0.632** | **0.596** | **0.601** | **0.660** | **0.655** | **0.563** | **0.668** | **0.625** |

Table 4: Realism role-play evaluator performance (real recording test set). We report Pearson correlation coefficients for all dimensions. The Avg. column is the average of all dimensions.

| Config. | Model | Prosodic Dynamics | | | Emotional Expressiveness | | | Character Consistency | | Contextual Relevance | | Avg. |
|---------|-------|-------|--------|--------|----------|-----------|------------|----------|--------|-------|--------|------|
| | | Pitch | Rynthm | Stress | Accuracy | Intensity | Transition | Identity | Traits | Local | Global | |
| Zeroshot | Gemini2.5Pro | 0.159 | 0.070 | 0.103 | 0.070 | 0.052 | 0.096 | 0.092 | 0.077 | 0.075 | 0.028 | 0.082 |
| | GPT4o-audio | 0.067 | -0.013 | 0.035 | 0.013 | 0.042 | 0.036 | -0.004 | 0.130 | 0.065 | 0.018 | 0.039 |
| | KimiAudio | -0.115 | -0.039 | -0.116 | -0.020 | -0.122 | -0.080 | **0.114** | -0.032 | -0.137 | -0.080 | -0.063 |
| | Qwen2.5Omni | 0.078 | 0.071 | 0.043 | 0.074 | 0.047 | 0.045 | -0.082 | 0.076 | 0.102 | 0.098 | 0.055 |
| | Qwen2Audio | 0.055 | 0.057 | 0.088 | 0.072 | 0.077 | 0.057 | -0.006 | -0.040 | 0.050 | 0.111 | 0.052 |
| FewShot | Gemini2.5Pro | **0.319** | 0.179 | 0.263 | 0.111 | 0.078 | 0.184 | -0.001 | 0.118 | 0.216 | 0.074 | 0.166 |
| | GPT4o-audio | 0.299 | 0.193 | 0.222 | 0.121 | **0.377** | 0.344 | -0.013 | **0.248** | **0.301** | 0.168 | 0.226 |
| Finetune | DRAME-Eval | 0.288 | **0.270** | **0.362** | **0.277** | 0.331 | **0.397** | -0.034 | 0.077 | 0.217 | **0.279** | **0.247** |

decoding-based regression methods (Song & Bahri, 2025; Xiao et al., 2025; Wang et al., 2025b; Lukasik et al., 2024). We apply LoRA fine-tuning (Hu et al., 2022). All implementation details, including optimizer settings, training schedule, and sampling policy, are provided in Appendix L.3.

We further conduct ablation studies in Appendix L.4, examining: the choice of different base audio LLMs and full-parameter fine-tuning versus LoRA adaptation. These analyses highlight the robustness of our sampling–expectation training strategy and clarify the trade-offs in model design.

**Evaluation Metrics for Role-play Evaluation Model.** For most dimensions with 1-5 scale, we report Pearson correlation. For *Content Pass* estimation, we report accuracy.

## 5.2 DRAME-EVALBENCH EXPERIMENTAL ANALYSIS

**Archetype Evaluation.** The results in Table 2 reveal clear distinctions across evaluation settings. In the zero-shot condition, proprietary ALLMs such as Gemini2.5Pro and GPT4o-audio substantially outperform public models, achieving strong *content pass* accuracy (∼90%) and moderate correlations with other dimensions(0.43–0.48). By contrast, public models exhibit limited predictive ability, with correlations often below 0.2. Few-shot prompting provides modest gains for proprietary models, improving both accuracy and correlation (e.g., Gemini2.5Pro rises from 0.480 to 0.493 average correlation). The most significant improvement arises from finetuning: DRAME-Eval achieves an average correlation of 0.629, marking a large margin over zero-shot/few-shot models and demonstrating the effectiveness of task-specific supervision.

**Realism Evaluation.** Table 3 extends the analysis to fine-grained realism evaluation. Similar to archetype results, proprietary ALLMs lead the zero-shot setting, with Gemini2.5Pro obtaining an average correlation of 0.390, compared to 0.284 for GPT4o-audio. Public models remain weak baselines. Few-shot prompting consistently improves performance: Gemini2.5Pro gains +0.042 and GPT4o-audio gains +0.083 over their zero-shot averages, showing the benefit of in-context supervision. Once again, finetuning is most effective: DRAME-Eval achieves 0.625 average

Table 5: Speech-DRAME benchmark for archetype role-play. Benchmark results are calculated based on `DRAME-Eval` discussed in Sec. 5.1. `G` and `Q` in the cascaded type refer to `Gemini2.5Pro` and `Qwen3-30B-Instruct`, respectively. Please refer to Sec. 6 for detailed models.

| Type | Model | Content Pass Rate | Audio Quality | Human Likeness | Approperiateness | Avg. |
|---|---|---|---|---|---|---|
| End-to-end | GPT-4o-realtime | 78.9% | 2.83 | 2.27 | 2.70 | 2.60 |
| | KimiAudio | 66.8% | 2.93 | 2.63 | 2.63 | 2.73 |
| | GLM4-Voice | 88.5% | 3.20 | 2.70 | 2.95 | 2.95 |
| | Qwen2.5Omni | 53.8% | 2.42 | 2.16 | 2.26 | 2.28 |
| | Qwen3Omni | 57.2% | 2.48 | 2.20 | 2.34 | 2.34 |
| | StepAudio2Mini | 90.5% | 3.31 | 2.84 | 2.96 | 3.04 |
| Cascaded | Q-F5TTS | 92.3% | 3.29 | 2.73 | 2.96 | 2.99 |
| | Q-IndexTTS2 | **94.7%** | 3.31 | 2.74 | **3.05** | 3.03 |
| | Q-MaskGCT | 91.4% | 3.18 | 2.65 | 2.95 | 2.93 |
| | Q-Vevo1.5 | 79.5% | 2.98 | 2.54 | 2.72 | 2.75 |
| | Q-CosyVoice2 | 94.2% | **3.32** | 2.73 | 3.00 | 3.02 |
| | G-F5TTS | 91.7% | 3.31 | 2.80 | 2.99 | 3.03 |
| | G-IndexTTS2 | 94.3% | 3.35 | **2.81** | **3.05** | **3.07** |
| | G-MaskGCT | 92.5% | 3.24 | 2.77 | 3.00 | 3.00 |
| | G-Vevo1.5 | 72.0% | 2.81 | 2.40 | 2.55 | 2.59 |
| | G-CosyVoice2 | 93.4% | 3.34 | 2.77 | 3.01 | 3.04 |

correlation, higher than the best few-shot results. Importantly, the gains are consistent across all 10 realism dimensions, confirming that the sampling–expectation training strategy generalizes beyond simple classification to nuanced paralinguistic cues.

**Real Recording Test Set.** The real-recording evaluation in Table 4 is notably more challenging, with all models suffering substantial performance drops. Proprietary models in the zero-shot setting achieve only ∼0.04–0.08 average correlation, and public models often fall below zero. Few-shot prompting yields small but noticeable improvements, especially for `GPT4o-audio` (0.226 avg.), suggesting that exemplars partially mitigate domain mismatch between synthetic and real human speech. Interestingly, while finetuning achieves the highest overall average (0.247), its relative advantage over few-shot prompting is narrower than in the synthetic setting, and some dimensions (e.g., *Character Identity*) remain weak. We emphasize that this observed gap between synthetic and real recordings is not merely an artifact but an intended feature of Speech-DRAME, designed to expose the mismatch and drive future research into bridging synthetic-to-real generalization.

**Overall.** Across both archetype and realism evaluations, we observe three consistent trends: (i) zero-shot performance is limited, particularly for open-source models, (ii) few-shot prompting offers moderate but consistent gains, especially for proprietary ALLMs, and (iii) finetuning with task-specific supervision (`DRAME-Eval`) delivers the largest improvements, achieving state-of-the-art correlations across nearly all dimensions. Nevertheless, results on the real-recording test set underscore the gap between synthetic and real human role-play scenarios, motivating future work in data diversification and domain adaptation.

# 6 DRAME-ROLEBENCH

Based on pre-trained `DRAME-Eval`, we establish the DRAME-RoleBench to evaluate SFMs' role play ability. For archetype evaluation, 1,250 scenarios are constructed follows the same data construction pipeline as the **DRAME-EvalBench**. For realism evaluation, we reuse 2,000 scenarios in the **DRAME-EvalBench** test set. We follow the same prompt strategy as outlined in Appendix D.2.

For comprehensive evaluation, we consider a comprehensive set of candidate models, including six end-to-end models: `GPT-4o-realtime` (OpenAI, 2025b), `KimiAudio` (Ding et al., 2025), `GLM4-Voice` (Zeng et al., 2024), `Qwen2.5Omni` (Xu et al., 2025a), `Qwen3Omni` (Xu et al., 2025b), `StepAudio2Mini` (Wu et al., 2025); and ten cascaded models by combining two widely-used textual LLMs (i.e., `Gemini2.5Pro` (Comanici et al., 2025) and `Qwen3-30B-Instruct` (Yang et al., 2025)) and five recent TTS models (i.e., `F5TTS` (Chen et al., 2024c), `IndexTTS2` (Zhou et al., 2025b), `MaskGCT` (Wang et al., 2025d), `Vevo1.5` (Zhang et al., 2025b), and `CosyVoice2` (Du et al., 2024b)).

**Archetype evaluation.** On archetype role-play (Table 5), cascaded pipelines consistently outperform end-to-end systems. The strongest cascaded systems, particularly `G-IndexTTS2` and `Q-IndexTTS2`, achieve both high content pass rates (above 94%) and balanced scores across audio quality, human likeness, and appropriateness. By contrast, end-to-end models such as `StepAudio2Mini` and `GLM4-Voice` perform competitively in terms of content coverage (above

Table 6: Speech-DRAME benchmark for realism role-play. Benchmark results are calculated based on `DRAME-Eval` discussed in Sec. 5.1. `G` and `Q` in the cascaded type refer to `Gemini2.5Pro` and `Qwen3-30B-Instruct`, respectively. Please refer to Sec. 6 for detailed models.

| Type | Model | Prosodic Dynamics | | | Emotional Expressiveness | | | Character Consistency | | Contextual Relevance | | Avg. |
|---|---|---|---|---|---|---|---|---|---|---|---|---|
| | | Pitch | Rynthm | Stress | Accuracy | Intensity | Transition | Identity | Traits | Local | Global | |
| End-to-end | GPT-4o-realtime | 3.20 | 3.27 | 3.31 | 3.32 | 3.11 | 3.02 | 3.25 | 3.32 | 3.10 | 3.33 | 3.22 |
| | KimiAudio | 2.94 | 3.02 | 3.04 | 3.06 | 2.84 | 2.76 | 3.03 | 3.08 | 2.90 | 3.09 | 2.98 |
| | GLM4-Voice | 3.23 | 3.30 | 3.34 | **3.34** | **3.12** | **3.03** | 3.24 | 3.32 | 3.11 | 3.35 | **3.24** |
| | Qwen2.5Omni | 3.23 | **3.33** | 3.34 | 3.33 | **3.12** | 3.00 | **3.26** | 3.28 | **3.12** | **3.34** | **3.24** |
| | Qwen3Omni | 3.02 | 3.07 | 3.13 | 3.12 | 2.91 | 2.83 | 3.06 | 3.16 | 2.89 | 3.15 | 3.03 |
| | StepAudio2Mini | 3.00 | 3.06 | 3.07 | 3.05 | 2.81 | 2.73 | 3.00 | 3.01 | 2.97 | 3.11 | 2.98 |
| Cascaded | Q-F5TTS | 3.23 | 3.27 | 3.33 | 3.29 | 3.06 | 2.96 | 3.20 | 3.28 | 3.06 | 3.29 | 3.20 |
| | Q-IndexTTS2 | 3.26 | 3.30 | **3.36** | 3.32 | 3.11 | 3.01 | 3.23 | **3.33** | 3.10 | 3.33 | 3.23 |
| | Q-MaskGCT | 2.60 | 2.64 | 2.70 | 2.70 | 2.48 | 2.43 | 2.67 | 2.82 | 2.53 | 2.70 | 2.63 |
| | Q-Vevo1.5 | 2.93 | 2.96 | 3.05 | 3.00 | 2.75 | 2.66 | 2.93 | 3.00 | 2.81 | 3.01 | 2.91 |
| | Q-CosyVoice2 | 3.14 | 3.20 | 3.26 | 3.24 | 3.02 | 2.92 | 3.16 | 3.25 | 3.01 | 3.25 | 3.15 |
| | G-F5TTS | 3.23 | 3.29 | 3.31 | 3.27 | 3.10 | 3.01 | 3.25 | 3.27 | 3.10 | 3.30 | 3.21 |
| | G-IndexTTS2 | **3.24** | 3.30 | 3.32 | 3.27 | 3.10 | 3.01 | 3.25 | 3.27 | 3.10 | 3.30 | 3.22 |
| | G-MaskGCT | 3.21 | 3.28 | 3.30 | 3.27 | 3.08 | 2.98 | 3.24 | 3.25 | 3.09 | 3.29 | 3.20 |
| | G-Vevo1.5 | 2.97 | 3.00 | 3.04 | 3.02 | 2.83 | 2.74 | 3.00 | 2.99 | 2.86 | 3.04 | 2.95 |
| | G-CosyVoice2 | 3.18 | 3.25 | 3.27 | 3.25 | 3.06 | 2.97 | 3.22 | 3.25 | 3.05 | 3.27 | 3.18 |

88%), but their expressive scores remain slightly lower than their cascaded counterparts. This gap suggests that separating reasoning (LLM) from speech rendering (TTS) remains advantageous when archetype fidelity is emphasized. Nevertheless, the fact that top end-to-end systems already surpass 3.0 in average rating indicates rapid closing of the performance gap.

**Realism evaluation.** Realism role-play (Table 6) presents a different landscape. Here, performance differences between end-to-end and cascaded systems are less pronounced. Several end-to-end models, such as `GLM4-Voice` and `Qwen2.5Omni`, reach average scores of 3.24, matching or even exceeding the best cascaded pipelines. Importantly, realism scores are more evenly distributed across prosodic, emotional, character, and contextual dimensions, with most models clustering between 2.9–3.2. This convergence reflects the difficulty of realism-style evaluation: since the data preparation emphasized contrast with authentic human speech, models are judged under similar conditions, leading to smaller observed gaps.

**Human alignment.** We further conducted small-scale human studies with 10% of the benchmark data to verify the reliability of the benchmark. Archetype evaluation achieves a Spearman correlation of 0.706 between human judgments and `DRAME-Eval`, validating that the automatic evaluator closely tracks human preferences in this setting. Realism evaluation, however, achieves a more modest Spearman correlation of 0.375. This lower alignment is expected, since realism data construction did not prioritize fine-grained model separation but instead emphasized distinguishing human versus synthetic speech.[8] As a result, all models occupy a relatively narrow performance band, reducing sensitivity to model-level differences. We view this as evidence that realism evaluation highlights a harder, less saturated frontier where both SFMs and SEMs require further advances.

Overall, cascaded pipelines retain an edge in archetype scenarios, especially in content robustness and expressive fidelity, but end-to-end models are catching up quickly. In realism scenarios, the field remains unsolved: existing systems exhibit similar performance bands and evaluators show weaker alignment with human judgments, underscoring the need for improved realism-grounded data and evaluation. Together, these findings establish **DRAME-RoleBench** as both a reliable testbed for archetype fidelity and a challenging benchmark for realism fidelity.

## 7 CONCLUSION

We presented **Speech-DRAME**, a unified framework for assessing spoken role-play through two complementary lenses: *Archetype Evaluation* (top-down) and *Realism Evaluation* (bottom-up, human-grounded). Our human-annotated **DRAME-EvalBench** enables training and testing of SEMs, and our fine-tuned **DRAME-Eval** substantially outperforms zero-/few-shot ALLM judges. Building on this, **DRAME-RoleBench** provides system-level comparison across various SFMs, revealing a consistent edge for cascaded pipelines on archetypes and a tighter spread under realism,

---

[8]We view this as evidence that realism presents a harder, less-saturated frontier. Rather than a limitation of DRAME-Eval, this highlights the challenge of designing evaluators that capture nuanced human perception, which motivates future work.

where real human performance evaluation remain challenging. Together, these resources establish a reproducible, analyzable foundation for spoken role-play evaluation.[9]

# 8 ETHICS STATEMENT

All data collection and annotation in this work followed strict ethical standards. Our bilingual datasets were constructed exclusively from legally accessible sources, including licensed media, open-source corpora, and original recordings conducted with informed consent. Annotators were recruited with appropriate compensation and provided with clear instructions, including the right to withdraw at any point. To minimize risks, no personally identifiable information was included in the released data, and all recordings were anonymized before annotation. We ensured cultural and linguistic diversity in annotator pools to mitigate potential bias across Mandarin and English role-play scenarios. While this study did not require formal IRB approval, we adhered to best practices in human-subject research, including transparency in task design and careful consideration of annotator well-being. Full details of our annotation protocol and ethical safeguards are presented in Appendix I.

# 9 REPRODUCIBILITY STATEMENT

We emphasize reproducibility as a core design principle of Speech-DRAME. Specifically, we detail annotation rubrics and protocols, with rejection rules and scoring scales for both strategies

Further, to enable transparent and verifiable research, we will release the following artifacts after the review period:

- Bilingual human-annotated datasets for both Archetype and Realism Evaluation, including train/test splits and benchmark subsets.

- 

- Training recipes and evaluation code for Speech Evaluation Models (SEMs), including hyperparameters and model checkpoints.
- Benchmarking pipeline, covering both cascaded and end-to-end speech foundation models, with standardized prompting templates and evaluation scripts.

Our experiments were conducted on widely available GPUs (NVIDIA H100). Random seeds are fixed across all runs to ensure consistency. By releasing datasets, models, and code, we aim to make all major results in this paper fully reproducible and facilitate future extensions of our framework.

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

# Appendices

APPENDIX CONTENTS

## A  USE OF LARGE LANGUAGE MODELS

Large language models (LLMs) and audio-capable LLMs were used in four capacities in this work:

1. **Data generation:** LLMs were employed to expand prompts (e.g., character profiles, scene descriptions) and to produce negative samples in realism evaluation. All outputs were screened by human annotators to ensure validity and reduce bias.

2. **Baseline evaluators:** State-of-the-art audio LLMs (e.g., `GPT4o`, `Gemini2.5Pro`, `Qwen2.5Omni`) were included as zero-shot judges in our evaluation benchmark, enabling direct comparison with human-aligned SEMs.

3. **Annotation design and task construction:** `Gemini2.5Pro` was used to assist with building instructions and summarizing story contexts. Drafted materials were subsequently refined by the authors and validated by human annotators.

4. **Manuscript preparation:** LLMs were used for polishing, proofreading, and improving clarity of the paper. All conceptual design, analysis, and substantive writing remain the work of the authors, with LLMs serving only as language assistants.

Proprietary LLM outputs were used in a limited way (e.g., instruction building, story summarization, or baseline evaluations) and only for internal research purposes. Released datasets in Speech-DRAME are constructed from open-source models, licensed corpora, and original human recordings.

## B  THEORETICAL MOTIVATION FOR DUAL EVALUATION

Our distinction between *Archetype Evaluation* and *Realism Evaluation* is not merely pragmatic, but is grounded in established theories of social identity, communication, and performance. The two perspectives capture complementary dimensions of how humans perceive and evaluate roles in interaction.

**Archetypes as Collective Schemas** The notion of archetypes originates from Jungian psychology, where they represent recurrent, culturally embedded character prototypes in the collective unconscious (Jung, 2014). In media and performance studies, archetypes have been shown to guide audience expectations, providing top-down interpretive frames for evaluating whether a character "fits" within familiar categories such as hero, mentor, or trickster (Campbell, 2008). Archetype-based reasoning allows for scalable, general-purpose judgments because it leverages culturally shared schemas rather than idiosyncratic details.

**Realism as Situated Performance** In contrast, realism highlights the fine-grained enactment of roles in situated interaction. Goffman's dramaturgical sociology emphasizes that everyday social life is itself a performance, where believability depends on subtle paralinguistic and pragmatic cues such as prosody, hesitation, and emotional shading (Goffman, 1949). Similarly, interactional sociolinguistics has demonstrated how micro-level features like intonation, pacing, and turn-taking patterns shape the perception of authenticity in communication (Gumperz, 1982). Realism Evaluation thus anchors role-play assessment in bottom-up sensitivity to authentic delivery, beyond schematic expectations.

**Complementarity of the Two Perspectives** Performance theory provides a unifying lens for these dual dimensions. Schechner's view of performance as both "restored behavior" (scripted, repeatable structures) and "living process" (situated enactment) (Schechner, 2017) underscores the need to consider both archetypal scripts and enacted realism. Likewise, sociolinguistic theories of identity recognize both *ascribed categories* (archetypes imposed by social convention) and *performed identities* (emergent in interaction) (Bucholtz & Hall, 2005).

Accordingly, Archetype Evaluation and Realism Evaluation are best viewed as complementary:

- Archetype Evaluation provides macro-level judgments grounded in socially shared schemas, enabling efficient and broad comparisons.

- Realism Evaluation captures micro-level cues of authenticity, ensuring sensitivity to nuanced delivery and interactional fit.

Together, they mirror the dual nature of human role assessment, balancing scalability and comparability with depth and nuance. This theoretical grounding reinforces the design of Speech-DRAME as a framework that does not privilege one perspective over the other, but instead integrates both to reflect the full spectrum of how roles are evaluated in speech-based interaction.

## C   EXTENSION TO MULTI-TURN ROLE-PLAY EVALUATION

Our main formulation (Section 3.1) focuses on the *single-turn* setting, where each response $\boldsymbol{S}_r$ is generated and evaluated independently. This simplification enables clearer analysis of speech role-play quality and aligns with our bilingual annotation protocol. Nevertheless, the same formulation extends naturally to multi-turn dialogues.

**Multi-turn generation.** Consider a dialogue of $T$ turns, with context $\{\boldsymbol{C}_{\text{speech}}^{(t)}\}_{t=1}^{T}$, profile $\boldsymbol{C}_{\text{profile}}$, and scene $\boldsymbol{C}_{\text{scene}}$. At each turn $t$, the SFM produces

$$\boldsymbol{S}_r^{(t)} \;=\; \text{SFM}\big(\boldsymbol{C}_{\text{speech}}^{(t)},\, \boldsymbol{C}_{\text{profile}},\, \boldsymbol{C}_{\text{scene}}\big). \tag{4}$$

**Multi-turn evaluation.** Each generated speech segment is scored by an SEM:

$$\boldsymbol{E}_r^{(t)} \;=\; \text{SEM}\big(\boldsymbol{S}_r^{(t)},\, \boldsymbol{C}_{\text{profile}},\, \boldsymbol{C}_{\text{scene}}\big). \tag{5}$$

This yields a sequence of per-turn scores $\{\boldsymbol{E}_r^{(1)}, \ldots, \boldsymbol{E}_r^{(T)}\}$.

**Dialogue-level aggregation.** A dialogue-level evaluation can be obtained by applying an aggregation operator $\Psi$ over turn-level scores:

$$\bar{\boldsymbol{E}}_r \;=\; \Psi\big(\boldsymbol{E}_r^{(1)}, \ldots, \boldsymbol{E}_r^{(T)}\big), \tag{6}$$

where $\Psi$ may be a simple mean, a recency-weighted average, or a learned temporal model that captures consistency across turns. The choice of $\Psi$ reflects the intended evaluation focus, e.g., stability of persona traits, adaptation to scene progression, or coherence across multiple speakers.

**Discussion.** While our released dataset and experiments concentrate on single-turn evaluation, the modular nature of the formulation ensures that extending to multi-turn is straightforward. Future work can leverage $\Psi$ to study richer aspects of speech role-play, such as persona persistence, interaction dynamics, and role consistency across extended dialogues.

# D  IMPLEMENTATION DETAILS FOR ARCHETYPE EVALUATION TASKS

This appendix first discusses the summary of task preparation steps for archetype evaluation. Then we provide concrete examples and extended discussion of the archetype evaluation annotation pipeline that are omitted in the main text for clarity.

## D.1  ARCHETYPE EVALUATION TASK PREPARATION

Archetype Evaluation follows a top–down perspective rooted in sociology and role theory, where social behavior is interpreted through shared stereotypes and generalized expectations (Goffman, 1949; Campbell, 2008). This strategy abstracts role-play into broad and easily recognizable archetypes, enabling scalable evaluation of whether a generated response aligns with salient role features.

**Role dimensions.** We operationalize archetypes through five principal categories, each representing a stereotypical dimension of role identity or behavior, including (1) **Occupation / Functional Role (Modern Daily Life)** (e.g., a firefighter rescuing a trapped civilian). (2) **Occupation / Functional Role (Fictional or Fantasy)** (e.g., a martial arts cult leader confronting orthodox rivals). (3) **Social Identity** (e.g., a doting parent worrying about a child, or a "Buddhist youth" eschewing competition). (4) **Named / Specific Characters** (e.g., Sun Wukong mocking celestial authorities in *Journey to the West*). (5) **Personality Traits** (e.g., "hot-blooded" in a competitive setting, or "complaining" in response to poor service). (6) **Relative Roles** (e.g., parent or stranger).

**Scene dimensions.** In addition to roles, we incorporate contextual triggers that anchor role-play in stereotypical situations. A central focus is on **everyday external events or state changes** (e.g., receiving a promotion, being caught in gossip), which provide concrete scenarios for testing archetypical responses.

**Exclusion of fine-grained controls.** We deliberately excluded explicit emotional states, prosodic instructions, or linguistic constraints. These dimensions are better suited to realism-oriented or style-controlled evaluation, whereas archetype evaluation benefits from concrete, stereotype-level scenarios that sharpen judgments and improve human agreement.

**Pipeline overview.** At a high level, archetype evaluation data are generated through a three-stage pipeline: (1) selecting keywords that specify role and scene; (2) expanding them into full prompts via structured templates; and (3) rendering prompts into speech with contemporary TTS systems. This modular pipeline ensures both scalability and reproducibility, while keeping prompts vivid enough for reliable judgments. (4) Based on the generated data (paired scene and synthesized speech), expert human annotators are recruited for detailed annotation on semantic check (i.e., *Content Pass*) and three different spoken role-play dimensions (i.e., *Audio Quality*, *Human-likeness*, *Appropriateness*). Implementation details, including template design, alternative prompting strategies, and annotation guidelines, are provided in the following subsections of this Appendix D.

Overall, archetype evaluation emphasizes breadth and recognizability over subtlety. By grounding role-play in socially recognizable categories and stereotypical situations, it provides a practical balance between scalability and interpretability, complementing the more fine-grained realism evaluation described in Section 3.2.

## D.2  ROLE AND SCENE CATEGORIES

We constructed archetype evaluation prompts by combining role categories with contextual scenes. The following illustrates the core categories with representative examples.

**Roles.**

- **Occupation / Functional Role in Modern Daily Life.** Example:
  - *You are a firefighter. Thick smoke fills the building, and you discover a trapped civilian. You say:*
- **Occupation / Functional Role in Fictional or Fantasy Settings.** Example:
  - *You are the leader of a dark cult in a martial arts fantasy world. The righteous alliance has come to challenge you outside your stronghold. You say:*

- **Social Identity.** Example:

  - *You are an overprotective parent. Your child has not come home late at night, and you cannot reach them by phone. You say:*
  - *You are a laid-back "go-with-the-flow" friend. Your colleague excitedly tells you about a highly competitive promotion and urges you to go for it. You say:*

- **Named / Specific Characters.** Example:

  - *You are Sherlock Holmes. In the courtroom, your rival lawyer mocks your deductions in front of the jury. You remain unimpressed. You say:*

- **Personality Traits.** Example:

  - *Your personality is defined as hot-blooded. Your team is losing a game, and in the final moments you must deliver the decisive score. You say:*
  - *Your personality is defined as constantly complaining. You are dining at a restaurant where the food is bland and the service disappointing. You say:*

- **Relative Roles.** Example:

  - *You are Parent. Your child returns home late at night after not answering calls. You say:*
  - *You are Stranger. You see a stranger's bag is unzipped and items are about to fall out, so you decide to alert them. You say:*

**Scenes.**

- **Everyday External Event / State Change.** Example:

  - *Your boss calls you into the office and tells you that because of your excellent performance, you are being promoted with a raise. You return to your desk, unable to hide a smile. You say:*
  - *You overhear coworkers gossiping and realize you have become the subject of their rumors. Your face turns red. You say:*

**Excluded Dimensions.** We initially considered more fine-grained control dimensions, but excluded them to preserve the archetype-level focus. Examples of excluded dimensions include: *Emotional State & Expression, Single Emotion Prompts, Explicit User Input / Assistant Output structures, Emotional Transition / Fluctuation, Voice Style & Prosody (pacing, intonation, vocal timbre, expressive style, accent/dialect), and Content / Interaction Patterns (task/topic controls, specific linguistic structures, or speech production challenges).*

These dimensions are more closely aligned with explicit instruction-following or style transfer rather than broad stereotype judgment, and they reduced clarity in human evaluations.

**The Choice of Prompts.** We also experimented with broader, less specific templates that allowed more freedom for generative models but produced less discriminative evaluation results:

- *You are [role]. You usually say:*
- *You are a {{ role }} in the {{ universe }}. Someone approaches you for help. You say:*

While LLMs often responded fluently to these broad templates, human annotators found judgments to be more ambiguous and tolerant, making scores less sharp. The final design therefore emphasizes more concrete, vivid prompts, as elaborated in the above examples in Appendix D.2.

### D.3 PIPELINE ILLUSTRATION

**Text generation.** We followed a three-step expansion: keyword → scene description → full prompt with template. Some examples are shown as follows:

- Occupation: firefighter → "smoke fills the building, you discover a trapped civilian" → *You are a firefighter. Smoke fills the building, and you discover a trapped civilian. You say:*

- Fictional role: warrior in a fantasy world → "you witness bullies harassing a weaker person" → *You are a warrior in a martial arts fantasy world. You see bullies harassing a weaker person. You say:*

- Event: promotion → "your boss calls you in to announce a raise" → *Your boss calls you into the office and announces you are being promoted with a raise. You return to your desk, smiling uncontrollably. You say:*

All steps are automatically processed by `Gemini2.5Pro` (Comanici et al., 2025), but each output is manually verified to ensure high quality for downstream usage. We demonstrate an example for Roles: Occupation / Functional Role in Fictional or Fantasy as follows:

---

**Step 1: Generate fictional universes**

```
Instruction:  Generate {{n_samples}} common fictional or
cinematic universes as keywords, and output them in JSONL
format.  The universes should cover a wide range of genres
(e.g., cyberpunk, martial arts wuxia, urban fantasy, high
fantasy magic), avoiding redundancy or over-concentration.

Output format:
{"universe_en": "..."}
{"universe_en": "..."}
```

---

**Step 2: Generate occupations/roles**

```
Instruction:  As a rigorous character designer, prepare
{{n_samples}} occupation or functional role types for the
given universe.

Examples:
- Fantasy Magic:  mage, knight, dragon, bard, saint, demon,
demon lord, elf, dwarf, goblin
- Science Fiction:  interstellar explorer, AI robot, cyborg,
genetic engineer, space pirate, colonial governor
- Post-apocalyptic:  survivor, scavenger, raider, mutant,
dungeon guard, resistance member
- Cosmic Horror:  investigator, cultist, occult scholar,
apostle of the Old Gods, cursed individual, mad prophet

Output format:
{"role_en":"...", "universe_en":"..."}
{"role_en":"...", "universe_en":"..."}
```

---

**Step 3: Final prompt generation**

```
Instruction:  As a top-tier role-play scenario designer,
create concise and effective prompts for evaluating
voice-based role-play.  Each prompt must directly
activate the role's core personality and speaking style
without explicitly describing mannerisms (e.g., \gently,"
\decisively").

Requirements:
1.  Conciseness:  Each scenario_en must be minimal and focus
on the essential trigger.
2.  Personality activation:  The scene must elicit
```

```
distinctive identity, social status, and communication style
(tone, vocabulary, rhythm).
3.  Immersion:  Scenarios should create immediate role
immersion with tension or iconic role traits.
4.  Personality--scenario linkage:  Each prompt must include
a personality_analysis summarizing the role's dominant
traits, which align with the designed scenario.

Example output:

{"role_en": "Demon Lord", "universe_en": "Fantasy Magic",
 "personality_analysis": "arrogant, scornful, controlling",
 "scenario_en": "A captured human hero defiantly
     glares at you before your throne."}

{"role_en": "Demon", "universe_en": "Fantasy Magic",
 "personality_analysis": "cunning, patient, manipulative",
 "scenario_en": "You tempt a desperate mortal,
     promising to grant their wish."}
Final expansion for each role keyword ({{n_samples}} scenarios
per role):

{"role_en": "{{ keyword.role_en }}",
 "universe_en": "{{ keyword.universe_en }}"}
```

**Speech generation.** Prompts were rendered into audio with CosyVoice2-0.5B Du et al. (2024b). The prompts are selected from human-verified high-quality samples, where the selection is conducted from open-source datasets, such as LibriSpeech/Gigaspeech for English, and WenetSpeech for Chinese (Panayotov et al., 2015; Chen et al., 2021; Zhang et al., 2022).

### D.4 CANDIDATE MODELS FOR EVALUATION

We evaluate a diverse set of speech foundation models spanning both English and Mandarin. In total, we include 8 English and 7 Mandarin models, covering open-source and proprietary systems, and balancing end-to-end and cascaded architectures. Table 7 provides the full list of models.

Table 7: Candidate speech foundation models for evaluation across English and Mandarin. `G` and `Q` in for cascaed models refer to `Gemini2.5Pro` and `Qwen3-30B-Instruct`.

| Model | License / Availability | Architecture | Reference |
|---|---|---|---|
| **English Models (8)** | | | |
| GPT-4o-realtime | Proprietary | Unknown | (OpenAI, 2025b) |
| Doubao | Proprietary | Unknown | (ByteDance, 2025) |
| GPT-4o(Advanced Voice Mode) | Proprietary | Unknown | (OpenAI, 2025a) |
| GLM4-Voice | Open-source | End-to-end | (Zeng et al., 2024) |
| KimiAudio | Open-source | End-to-end | (Ding et al., 2025) |
| Qwen-2.5-Omni | Open-source | End-to-end | (Xu et al., 2025a) |
| Q-CosyVoice2 | Open-source | Cascaded | (Yang et al., 2025; Du et al., 2024b) |
| G-Cosyvoice2 | Proprietary + Open-source | Cascaded | (Comanici et al., 2025; Du et al., 2024b) |
| **Mandarin Models (7)** | | | |
| GPT-4o-realtime | Proprietary | Unknown | (OpenAI, 2025b) |
| Doubao | Proprietary | Unknown | (ByteDance, 2025) |
| GLM4-Voice | Open-source | End-to-end | (Zeng et al., 2024) |
| KimiAudio | Open-source | End-to-end | (Ding et al., 2025) |
| Qwen-2.5-Omni | Open-source | End-to-end | (Xu et al., 2025a) |
| Q-CosyVoice2 | Open-source | Cascaded | (Yang et al., 2025; Du et al., 2024b) |
| G-Cosyvoice2 | Proprietary + Open-source | Cascaded | (Comanici et al., 2025; Du et al., 2024b) |

**Collection for Proprietary Models.** For proprietary systems, we adopt two collection strategies. For GPT-4o-realtime, responses are obtained directly via the `gpt-4o-realtime` API. For Doubao and GPT-4o (Advanced Voice Mode), we collect outputs by recording from their official web interfaces. All recordings are captured through the interaction device itself, ensuring that no additional processing or distortion is introduced into the final audio.

**End-to-End Model Generation.** For end-to-end systems, the audio prompt is directly used as input. The system prompt is set to default values to ensure consistent speech generation quality.

**Cascaded Model Generation.** For cascaded systems, we only use Cosyvoice2 as the TTS model in our collection of data. We select `Qwen3-30B-Instruct` and `Gemini2.5Pro` as the text LLM candidates (Yang et al., 2025; Comanici et al., 2025).

During the generation, we adopt a retrieval-augmented generation (RAG) strategy that combines text LLMs with zero-shot TTS models:

- Step 1: We first build a speech database (LibriSpeech and GigaSpeech for English, Wenet-Speech for Mandarin (Panayotov et al., 2015; Chen et al., 2021; Zhang et al., 2022)). Each sample is annotated with Gemini-based speech captions describing salient acoustic attributes.

- Step 2: Given a character and scene description, the text LLM imagines a candidate caption and generates a corresponding transcript (see below prompts for the text LLM).

- Step 3: We retrieve acoustically similar samples from the database using `Qwen3-Embedding-7B` (Zhang et al., 2025c).

- Step 4: The retrieved samples guide the zero-shot TTS system, which synthesizes speech conditioned on both the transcript and matched acoustic profile.

---

**Caption Generation with Text LLMs for Cascaded SFM.**

```
Generate JSONL captions describing the vocal delivery for the
following scenes.  Match the style of the reference captions.

Each JSON line must contain:  'index', 'scene', 'caption'.

Reference Style:
{% for sample in sample_captions -%}
{{ sample.caption }}
{% endfor %}

Input Scenes:
{% for scenario in scenarios -%}
{"index": {{ scenario.index }}, "scene": "{{ scenario.scene
}}"}
{% endfor %}
```

---

**Response Generation with Text LLMs for Cascaded SFM.**

```
Generate the corresponding speech content for the following
scenarios.  Ensure the generated text is natural and
perfectly fits the context and tone of each scene.

Each JSON line must contain:  ìndex, ̀scene, ̀caption, ̀content.

Input Scenes:
{% for scenario in scenarios -%}
{"index": {{ scenario.index }}, "scene": "{{ scenario.scene
```

```
}}", "caption":  "{{ scenario.caption }}"}
{% endfor %}
```

# E   ANNOTATION GUIDELINES FOR ARCHETYPE EVALUATION

This appendix contains the full human annotation instructions used for archetype evaluation.

## E.1   INITIAL ASSESSMENT & REJECTION INSTRUCTIONS

- Focus on the first 30 seconds only.
- Reject if the generation is irrelevant, breaks role instructions, or contains severe defects.
- Reject if foreign-language content is included (except for widely recognizable words).
- Reject if the clip is too short (1–2s) or clearly breaks immersion.

Detailed case examples are provided in the full instruction document.

## E.2   RUBRIC RATING INSTRUCTIONS

We adopt three scales, each rated from 1 (worst) to 5 (best):

- **Audio Quality (No Context):** independence from content, focus on distortions or synthesis artifacts.
- **Human Likeness (No Context):** perceived naturalness and plausibility of human delivery.
- **Appropriateness (Context Considered):** tonal fit of the vocal performance to the described role and scene.

The complete rubrics with per-score definitions are reproduced below:

## E.3   COMMENTARY GUIDELINES AND CONFIDENCE RATINGS

Annotators are required to provide short, clear justifications:

- Comments are mandatory for Appropriateness (all scores).
- Comments for Audio Quality and Human Likeness are only required for scores $\leq 3$.
- Gender mismatches should be flagged in comments but not factored into rubric scoring.

Annotators also provide a confidence level (high, moderate, low) for their Appropriateness rating, reflecting familiarity with the role and scene. This provides additional granularity in analysis.

## E.4   FULL ANNOTATION GUIDELINE (ROLE-PLAY ARCHETYPE EVALUATION)

---

**Role-play Archetype Evaluation**

```
Goal.  The overall goal of the initial assessment for each
audio is to determine if any of the clips generated from each
model meet any criteria to receive a REJECTION (referred to
as ``content pass:  fail'' in our tasking interface).
```

---

**Initial Assessment & Rejection Instructions.** Listen to each audio *in full*, paying special attention to the following caveats and rejection criteria.

CAVEATS

- **30-second focus:** Assessment should focus on the first 30 seconds of the generation ONLY. Do not consider any content past 30s; disregard it entirely from your assessment and rubric ratings. (You may stop listening beyond 30s.)

- **Early breakdowns:** If repetition, restarts, or obvious breakdowns occur *before* 30s (or for total clips shorter than 30s), the clip may be rejected.

- **Excessively short clips:** If a clip is only 1–2 seconds, reject due to insufficient information for reliable rubric ratings.

- **Foreign language content:** If *any* part before 30s contains a foreign language (even a single word/short phrase) in otherwise English audio, reject the entire clip.
  *Exception:* Extremely common/recognizable borrowed words (e.g., "croissant") may pass. Niche items (e.g., "coq au vin") should be rejected.

REJECTION CRITERIA

1. **Instruction Failure.** Generated content does not adhere to the trait(s) or role described in the prompt ("question").

   - *Example (reject):* Prompt: "Your personality is sharp-tongued. Your friend asks about their new haircut." Audio says: "It looks great on you." ⇒ reject.
   - *Example (pass):* Same prompt; audio says: "It looks like your stylist put you through a wood chipper." ⇒ content pass = true.
   - *Fourth-wall breaks:* Reject any generation acknowledging the scenario or narrating its role (e.g., "I am a lawyer in a courtroom...") or speaking for imagined other parties (two-way dialogue narration).
   - *Stage-direction speak:* Phrases like "pats you on the back reassuringly" are grounds for rejection.
   - *Placeholders:* Very short/abstract placeholders like "nurse" or "concept" are acceptable; longer placeholders like "and you can fill in the details here" are rejectable.

2. **Irrelevance.** Generated content is entirely unrelated to the prompt.

   - *Example (reject):* Prompt: "You are a kindergarten teacher. You are helping a crying child." Audio: "I'd like to thank everyone for this prestigious award." ⇒ reject.
   - *Example (pass):* "Oh no. Are you hurt? Don't worry. We'll get you taken care of, little one." ⇒ content pass = true.

**Important:** Once an audio receives content pass = false, *do not* rate rubrics; proceed to the next clip.

**Rubric Rating Instructions (for content pass = true).** We shift focus from *what* was said to *how* it was said (vocal performance). Use gut intuition guided by rubric definitions. Score each audio *in isolation* (no ranking across clips).

COMMENTARY

- After each 1–5 rating, add a short comment explaining your score.

- For **Audio Quality** & **Human Likeness**: comment only if score ≤ 3; if 4 or 5, write "N/A".

- For **Appropriateness**: comment for *all* scores 1–5. Explain *why* the *delivery* was/was not a good match to the in-context traits (role & scene).

- Do *not* critique word choice or logic; past the initial rejection step, only the *performance* matters.

- Prefer brief, clear, non-technical comments; no need to quote transcripts.

GENDER MISMATCH FLAGGING (COMMENTS ONLY)

If the prompt indicates a specific gender, flag mismatches in comments (e.g., "GENDER MISMATCH"). Do *not* penalize rubric scores for gender mismatch; rate vocal performance as-is.

CONFIDENCE

Provide a confidence level (High / Moderate / Low) for your *Appropriateness* rating, reflecting your familiarity with the role/scene.

- Low confidence if unfamiliar with the role/scene (e.g., "Air Traffic Controller").

- High confidence if very familiar (e.g., "Grandmother").

- Lower confidence *will not* reflect poorly on annotators; it adds nuance to analysis.

RATING RUBRICS: GENERAL CAVEATS

- Judge within traditional *stereotypes* for the prompt; when in doubt between two scores, choose the *lower*.

- If any portion of the clip is ill-fitting, prefer the lower overall score; do not average across good and bad segments.

- Evaluate the *performance*, not the script content.

- Pronunciation errors: penalize only on **Human Likeness** if they significantly affect comprehension; do not affect **Appropriateness**.

AUDIO QUALITY (NO CONTEXT)

*Independently* rate technical clarity (ignore content and fit). After scoring Audio Quality, *disregard* its issues for the other two dimensions.

| Score | Definition |
|---|---|
| 5 Excellent | Clean, clear, high-quality throughout; no digital noise/glitches/issues. |
| 4 Good | Good overall; minor artifacts detectable on close listen (slight sharpness/hiss, etc.). |
| 3 Fair | Acceptable but obvious artifacts (buzz, echo, synthetic timbre) impacting clarity. |
| 2 Poor | Frequent/consistent glitches or severe issues; distracting metallic/clipping/harsh tones. |
| 1 Very Poor | Extremely poor; layered artifacts dominate; severe clipping/static/failures, hard to understand. |

HUMAN LIKENESS (NO CONTEXT)

Rate how likely an average listener would believe the clip is human-delivered (ignore Audio Quality issues and contextual fit).

| Score | Definition |
|---|---|
| 5 Definitely Human | Indistinguishable from human: rich intonation, natural pacing, consistent identity; organic pauses/breaths, logical inflection. |
| 4 Most Likely Human | Generally convincing; slight stiffness/over-even pacing or minor awkward pauses. |
| 3 Could be Human or AI | Mix of natural and unnatural traits; borderline cases with both good and bad sections. |
| 2 Mostly Likely AI | Clearly artificial patterns: choppy timing, missing pauses, flat/illogical inflections, forced breaths. |
| 1 Definitely AI | Rigid, machine-like delivery throughout; flat/monotone, "read-aloud" style, no natural rhythm. |

APPROPRIATENESS (CONTEXT CONSIDERED)

Rate *only* how well the *delivery* fits the role/scene described in the prompt (tonal fit). Ignore Audio Quality and Human Likeness issues while scoring this dimension.

| Score / Criteria | Definition |
|---|---|
| 5 Completely Appropriate | Fully fits BOTH role and specific scene; aligns with emotional stance, social dynamics, and situational events; strong immersion. |
| 4 Mostly Appropriate | Minor issues: role slightly off but scene reaction good; or scene slightly off but role good; or both slightly off with no major issues. Strong overall match with small flaws (e.g., intensity/subtlety). |
| 3 Adequately Appropriate | Generally acceptable for BOTH role and scene but lacking depth; noticeable inconsistencies. |
| 2 Slightly Appropriate | EITHER role *or* scene reaction is clearly not acceptable; weak alignment or mismatched/missing elements. |
| 1 Completely Inappropriate | BOTH role and scene are contextually mismatched; tone/urgency/status illogical; heavy misfit with expected alignment. |

# F IMPLEMENTATION DETAILS FOR REALISM EVALUATION TASKS

This appendix starts with a summary of the task preparation pipeline for our proposed realism evaluation. Then, we provides the detailed prompts and system setups for constructing the Realism Evaluation dataset described in Section 4.2.

## F.1 REALISM EVALUATION TASK PREPARATION

Realism Evaluation adopts a bottom–up perspective grounded in real human speech data, with the goal of capturing the nuanced delivery, prosody, and contextual fit that synthetic archetype prompts cannot fully represent. Rather than relying on stereotypes, this strategy leverages real-world recordings and carefully constructed contrasts, enabling fine-grained assessment of model performance under realistic conditions.

**Data from real human speech sources.** We construct the realism evaluation set primarily from natural speech material:

- **Ground-truth audio from TV and film.** Spoken segments are curated from publicly available media, ensuring coverage across diverse roles and communicative settings.
- **Retrieved character profiles.** Each audio segment is paired with a concise character description derived from the media context. The detailed prompt design for generating these character profiles is provided in Appendix F.2.
- **Generated local scenes.** We supplement retrieved character information with automatically generated local scene descriptions summarizing the surrounding plot or setting. Appendix F.3 includes representative prompts and generation details.

**Negative sample generation.** To stress-test evaluation models, we construct contrastive negative examples that deliberately mismatch role and scene information:

- **Negative local scenes.** For each original segment, we generate additional scene descriptions at different levels of mismatch, ranging from subtle alterations to fully unrelated scenarios. Appendix F.4 outlines the prompting strategies and example outputs.
- **Negative character profiles.** We also create role-profile mismatches by resynthesizing transcripts with diverse TTS systems, thereby breaking the link between authentic delivery and intended role. We explore both text-instruction-based prompting and comparisons across commercial and non-commercial TTS frameworks. Detailed model lists and parameter settings are documented in Appendix F.6.

Together, real human ground-truth segments and systematically constructed negatives form a balanced dataset that supports realism-oriented evaluation. The positives provide natural anchors of role-play performance, while the negatives introduce controlled mismatches that test whether evaluation models can distinguish real human speech from synthetic approximations. This design ensures

that realism evaluation emphasizes subtle paralinguistic and contextual cues, moving beyond intelligibility or content fidelity.

During annotation, raters are presented with either: (i) ground-truth stories, (ii) stories with negative local scenes, or (iii) stories with negative character profile. Annotators assign realism scores without being exposed to the underlying mismatch reasons. Instead, these reasons are provided only to the quality-assurance team, who use them as reference for internal validation and consistency checks. While the mismatch reasons may not be perfect, they help streamline quality control procedures and maintain reliability across the dataset.

## F.2 PROMPTS FOR CHARACTER PROFILE CREATION

We designed prompts to extract or generate concise role descriptions corresponding to characters in TV, film, or natural dialogue sources. These descriptions emphasize identity, relationship, and communicative goals.

---

**Example Prompt: Character Profile (Retrieval)**

```
You are a character-description assistant.

Task
I will give you a character name and the name of the show or
game they are from.  Your job is to check the internet and
return a concise description of the character that focuses on
how they behave when interacting with others.

Input
Character: {speaker}
Context: {show_or_game}

Description guidelines
- Focus specifically on the character's behavior and
personality, especially how they express themselves when
talking to others.
- Cover all aspects of how they interact with people, so
that their overall communication style and relationships are
clear.
- Do NOT mention the show or game name in the description.
- *Never* mention or speculate about accents.
- The profile should contain NO guessing words like "maybe",
"might", "could be", "likely", etc.
- The profile must be internally consistent, include no
contradictions.
- If the context is a game, do NOT mention anything about
interacting with a player.  Only describe how the character
interacts with other characters.
- Do NOT include any actor or voice actor information.  -
Lilith's example:  Lilith is known for her exceptionally
cold and emotionally detached demeanor.  Her interactions are
often characterized by a flat affect, precise articulation,
and a generally humorless approach.  She speaks in a
measured, almost clinical tone, rarely betraying strong
emotions.  When interacting with others, she is highly
intelligent and articulate, but also critical and often
condescending, especially towards those she deems less
```

```
intellectually gifted.  She has a knack for delivering
cutting remarks with a dispassionate facade, which can make
her intimidating and unsettling.  She seems incapable of
expressing warmth or affection in a conventional manner,
often making others feel uncomfortable or inadequate in her
presence.
- Use less than 100 words for profile description.

Return value (strict JSON, minified)
```json
"character":"character name","context":"show or game
name","full_profile":"full profile"

```
```

**Example Prompt: Character Profile (Reasoning)**

```
You are a character-reasoning assistant.

Task
Analyze the entire dialogue below and infer a profile
of Speaker {target_speaker}.  Draw on context from every
participant plus the emotion/audio captions, but focus your
profile solely on Speaker {target_speaker}.

Dialogue:
{dialogue_context}

Emotion caption for Speaker {target_speaker} (if available):
{emotion_caption}

Audio caption for Speaker {target_speaker} (if available):
{audio_caption}

Profile-writing guidelines
- Never mention or speculate about accents.
- Please use the name of character indicated from the
dialogue context, but if not, you must make a creative guess
based on the dialogue.  Do this for both speakers.
- Do NOT use generic names like "Speaker 0" or "Speaker 1".
- The profile should contain NO guessing words like "maybe",
"might", "could be", "likely", etc.
- The profile must be internally consistent, include no
contradictions.
- Ignore or omit any discussion of conversational transitions
or turn-taking.
- Where details are missing, make informed, creative
inferences to complete the portrait (see the "Lilith" example
you provided for the desired depth and tone).
- Lilith's example:  Lilith is known for her exceptionally
cold and emotionally detached demeanor.  Her interactions are
often characterized by a flat affect, precise articulation,
```

```
and a generally humorless approach.  She speaks in a
measured, almost clinical tone, rarely betraying strong
emotions.  When interacting with others, she is highly
intelligent and articulate, but also critical and often
condescending, especially towards those she deems less
intellectually gifted.  She has a knack for delivering
cutting remarks with a dispassionate facade, which can make
her intimidating and unsettling.  She seems incapable of
expressing warmth or affection in a conventional manner,
often making others feel uncomfortable or inadequate in her
presence.
- Use less than 100 words for profile description.

Return value (strictly minified JSON)
```json
{{"full_profile":"your profile text here", "speaker0":  "name
of speaker 0", "speaker1":  "name of speaker 1"}}
```
```

### F.3 PROMPTS FOR LOCAL SCENE GENERATION

To provide consistent evaluation context, we generate local scene descriptions summarizing the situational background. These prompts aim to capture the immediate narrative context without introducing extraneous detail.

**Example Prompt: Local Scene (Reasoning)**

```
Task:
Given the following inputs:
- Episode information from a TV show or movie
- A full transcript of a scene (either a dialogue or a
monologue)
- General background information about a specific character
- A selected span of three consecutive sentences spoken by
that character

Your goal is to analyze the localized story context
surrounding those three lines, focusing on the character's
intent, emotional state, and interaction dynamics.

First, integrate the episode information to understand the
broader plot, timeline, and narrative setup.  Then, use the
character's general background to ground your interpretation
of their behavior, personality traits, and emotional
expressions.  These two layers of context (episode-level
and character-level) should inform your understanding of the
local moment in the transcript.
Try to use the name refer in the character general
information instead of speaker 0 or speaker 1.  Even if no
information is inlcuded, simple use speaker WIHOUT number.

Inputs:  {{Character general information}}
{{Full transcript}}
{Character of interest}
{Three consecutive sentences spoken by that character}
```

```
Output Format (in a JSON format, with the following fields):
{{
"Immediate_Situation":  "Describe what is happening in the
scene just before and during these lines.  Include relevant
environmental or narrative context.",
"Character_Intention":  "What is the character trying to do
or communicate through these lines?",
"Emotional_Subtext":  "What emotions are being expressed,
suppressed, or shifted?",
"Emotion_Subtext_List":  "List the sequence of emotions
by their occurrences, separated by commas.  ONLY use
emotions from the following list: {'Happy', 'Sad', 'Angry',
'Fearful', 'Disgusted', 'Surprised', 'Neutral'}.",
"Character_Dynamics":  "How do these lines reflect or affect
the character's relationship with others in the scene?",
"Overall_Summary":  "The summarization of the above aspects."
}}

The user input is:
{user_input}
```

### F.4   PROMPTS FOR NEGATIVE SCENES

Negative scenes are created by altering or mismatching the original context at varying levels of severity. This section provides the prompting templates and representative outputs.

---

**Example Prompt: Negative Local Scene (Level0)**

```
You are given:
- A character profile describing personality, habits, and
tone.
- A speech transcript the character says.
- A real story where both the character and the transcript
fit naturally.

Your task is to generate a fake local story that maintains
surface plausibility but introduces a controlled mismatch.
Return a JSON object with the following fields:

- 'fake_story':  A new scene where the character and
transcript still appear naturally.
- 'reason':  A short explanation of why the transcript is
subtly off in this new scene.

Constraints:
- Do not modify the character profile or speech transcript.
- Do not repeat the transcript in your fake_scene_description.
- Do not introduce new audio events unless they are essential
to the mismatch.
- The mismatch should arise from one of the following:
situational logic, emotional tone, or stylistic clash ---
```

---

```
not from sound cues or transcript changes.

Keep character and transcript fitting in tone, but change the
local context so the transcript becomes subtly unnecessary,
exaggerated, or slightly illogical.

Inputs:
- Character Profile: {character_global_story}
- Transcript: {local_transcript}
- Real Story: {real_story}
```

**Example Prompt: Negative Local Scene (Level1)**

```
You are given:
- A character profile describing personality, habits, and
tone.
- A speech transcript the character says.
- A real story where both the character and the transcript
fit naturally.

Your task is to generate a fake local story that maintains
surface plausibility but introduces a controlled mismatch.
Return a JSON object with the following fields:

- 'fake_story': A new scene where the character and
transcript still appear naturally.
- 'reason': A short explanation of why the transcript is
subtly off in this new scene.

Constraints:
- Do not modify the character profile or speech transcript.
- Do not repeat the transcript in your fake_scene_description.
- Do not introduce new audio events unless they are essential
to the mismatch.
- The mismatch should arise from one of the following:
situational logic, emotional tone, or stylistic clash ---
not from sound cues or transcript changes.

The transcript and context still logically connect, but
the emotional tone is off (e.g., too intense, too calm, too
cheerful given the context).

Inputs:
- Character Profile: {character_global_story}
- Transcript: {local_transcript}
- Real Story: {real_story}
```

**Example Prompt: Negative Local Scene (Level2)**

```
You are given:
- A character profile describing personality, habits, and
tone.
- A speech transcript the character says.
- A real story where both the character and the transcript
fit naturally.

Your task is to generate a fake local story that maintains
surface plausibility but introduces a controlled mismatch.
Return a JSON object with the following fields:

- 'fake_story':  A new scene where the character and
transcript still appear naturally.
- 'reason':  A short explanation of why the transcript is
subtly off in this new scene.

Keep the scene and character presence, but make the
transcript's style feel unnatural for the character (e.g.,
too logical, too emotional, too formal, etc.).

Constraints:
- Do not modify the character profile or speech transcript.
- Do not repeat the transcript in your fake_scene_description.
- Do not introduce new audio events unless they are essential
to the mismatch.
- The mismatch should arise from one of the following:
situational logic, emotional tone, or stylistic clash ---
not from sound cues or transcript changes.

Inputs:
- Character Profile:  {character_global_story}
- Transcript:  {local_transcript}
- Real Story:  {real_story}
```

**Example Prompt: Negative Global Scene (Level0)**

```
You are given:
- A fixed transcript (the exact lines spoken --- do not
change this).
- The original character profile (describing personality,
tone, style).
- The real story context (scene and situation where the
transcript was originally spoken).

Your task is to generate a fake character profile and a
lightly modified scene context, such that the same transcript
is still plausibly spoken, but introduces a graded mismatch
in personality, tone, or identity.
```

```
Return a JSON object with the following fields:

- 'fake_profile':  A description of a new character who still
plausibly says the transcript, but whose identity or traits
introduce a mismatch.
- 'fake_story':  A version of the real story context adapted
to fit this new character.
- 'reason':  A short explanation of how the transcript
feels off given the new character profile (superficially or
fundamentally).

Change surface traits (e.g., gender, age, species, name)
while preserving emotional and stylistic alignment.  The
transcript sounds natural emotionally, but voice or social
expectations may cause slight dissonance.

Constraints (All Levels):
- Do not modify the transcript.
- Keep the character's presence plausible in the story.
- Only alter global identity traits --- avoid local scene
logic changes unless needed to accommodate the new character.
- Do not invent or rely on new audio events unless absolutely
essential.

Inputs:
- Character Profile:  {character_global_story}
- Transcript:  {local_transcript}
- Real Story:  {real_story}
```

**Example Prompt: Negative Global Scene (Level 1)**

```
You are given:
- A fixed transcript (the exact lines spoken --- do not
change this).
- The original character profile (describing personality,
tone, style).
- The real story context (scene and situation where the
transcript was originally spoken).

Your task is to generate a fake character profile and a
lightly modified scene context, such that the same transcript
is still plausibly spoken, but introduces a graded mismatch
in personality, tone, or identity.

Return a JSON object with the following fields:

- 'fake_profile':  A description of a new character who still
plausibly says the transcript, but whose identity or traits
```

```
introduce a mismatch.
- 'fake_story':  A version of the real story context adapted
to fit this new character.
- 'reason':  A short explanation of how the transcript
feels off given the new character profile (superficially or
fundamentally).

Introduce a tone or speaking style that clashes with the
transcript (e.g., stiff character says something casual,
aloof character says something emotional).  The mismatch is
noticeable but still believable.

Constraints (All Levels):
- Do not modify the transcript.
- Keep the character's presence plausible in the story.
- Only alter global identity traits --- avoid local scene
logic changes unless needed to accommodate the new character.
- Do not invent or rely on new audio events unless absolutely
essential.

Inputs:
- Character Profile:  {character_global_story}
- Transcript:  {local_transcript}
- Real Story:  {real_story}
```

**Example Prompt: Negative Global Scene (Level2)**

```
You are given:
- A fixed transcript (the exact lines spoken --- do not
change this).
- The original character profile (describing personality,
tone, style).
- The real story context (scene and situation where the
transcript was originally spoken).

Your task is to generate a fake character profile and a
lightly modified scene context, such that the same transcript
is still plausibly spoken, but introduces a graded mismatch
in personality, tone, or identity.

Return a JSON object with the following fields:

- 'fake_profile':  A description of a new character who still
plausibly says the transcript, but whose identity or traits
introduce a mismatch.
- 'fake_story':  A version of the real story context adapted
to fit this new character.
- 'reason':  A short explanation of how the transcript
feels off given the new character profile (superficially or
fundamentally).
```

```
   Create a character whose personality, values, or emotional
   posture fundamentally contradict the transcript.  Their
   saying it requires unusual circumstances (e.g., parody,
   breakdown, irony).

   Constraints (All Levels):
   - Do not modify the transcript.
   - Keep the character's presence plausible in the story.
   - Only alter global identity traits --- avoid local scene
   logic changes unless needed to accommodate the new character.
   - Do not invent or rely on new audio events unless absolutely
   essential.

   Inputs:
   - Character Profile:  {character_global_story}
   - Transcript:  {local_transcript}
   - Real Story:  {real_story}
```

## F.5 EXAMPLES OF REALISM ROLE-PLAY

To make the construction of the realism evaluation task concrete, we present a worked example that begins with a curated local scene and character profile, followed by the same transcript adapted into mismatched contexts. These mismatches appear either at the *local-scene* level or at both the *character profile and scene* level, thereby providing controlled degradations of realism.

We use the character Yunjin from *Genshin Impact* as a running case study, which is also included in our released dataset.[10] This example illustrates how alignment between a speaker's persona, local context, and delivery underpins realism, and how systematically breaking that alignment produces contrastive test cases.

**Curated Local Scene.** Yunjin is narrating a fierce battle through operatic singing. Her verses depict a hero charging into enemy lines, slaying the enemy commander, and securing peace. The atmosphere is filled with tension and intensity, highlighting the hero's extraordinary bravery and determination.

**Character Profile.** Yunjin is a passionate opera performer with deep knowledge of traditional singing art. Her lyrics vividly portray battlefield struggles and heroic deeds, conveying admiration for those who stand firm against overwhelming odds. She always seeks to inspire her audience with courage, justice, and hope, channeling her artistry into both entertainment and moral elevation.

**Original Transcript (English Translation).**

> **Yunjin:** Armored riders charge into the enemy lines, no longer entangled with the blue-clad foes.
> **Yunjin:** Deep amidst the horde of demons, the enemy commander is seized, his red robe cut down by the sword.
> **Yunjin:** A blade of frosty steel strikes fear into their hearts, calming the storms and securing peace.
> **Yunjin:** It is said,
> **Yunjin:** Though thousands of soldiers guard the pass, one hero holds it alone; even with blades upon their body, they remain undaunted.

**Negative Local Scenes.** To stress realism evaluation, we insert the same transcript into alternative backgrounds. These reveal how context-shifted delivery degrades narrative coherence:

---

[10]As acknowledged, the use of real-world recordings is conducted under legal and ethical review. All curated data will be released as part of our open-source package.

- **Level 0 (Mild Mismatch).** *Fake Scene:* A quiet teahouse where Yunjin casually chats with friends about recent weather. She suddenly delivers the heroic verses in full operatic style. *Reason:* The shift from light conversation to intense battle imagery feels abrupt and unsupported. The mismatch is mild but noticeable.
- **Level 1 (Moderate Mismatch).** *Fake Scene:* A teahouse with elderly patrons enjoying tea in a calm setting. Yunjin, dressed plainly, begins singing passionately. *Reason:* The intense tone clashes with the subdued environment, startling listeners. Her heroic lines sound out of place, generating tonal dissonance instead of inspiration.
- **Level 2 (Severe Mismatch).** *Fake Scene:* A modern café with soft jazz music and casual chatter. Yunjin sips a latte while reading a magazine, then recites the battle verses. *Reason:* The dramatic imagery of warriors and swords is wholly incompatible with the relaxed, contemporary café ambience. The mismatch is extreme, rendering the delivery absurd.

**Negative Character Profiles.** We replace Yunjin with new character profiles and scenes. Although the transcript remains the same, realism degrades because the delivery no longer matches the speaker's persona or situation.

- **Level 0 (Mild Mismatch).** *Fake Scene:* At a modern tech company's annual party, a shy programmer is asked to perform. Under pressure, he recites opera lines he once saw online. *Fake Profile:* Zhang Wei, an introverted software engineer with little artistic background, usually immersed in code and algorithms. *Reason:* The heroic verses sound alien coming from someone unfamiliar with opera or traditional culture. The clash produces awkward humor but still feels like a coping attempt.
- **Level 1 (Moderate Mismatch).** *Fake Scene:* In a futuristic laboratory, a robotic engineer celebrates a successful combat simulation. Overwhelmed with excitement, he blurts out the opera verses. *Fake Profile:* Xiao A, a calm and rational roboticist, normally concise and analytical, not prone to poetic expression. *Reason:* The dramatic, metaphorical verses conflict with his logical persona and sterile environment. The mismatch is sharper: the heroic rhetoric feels uncharacteristic and jarring.
- **Level 2 (Severe Mismatch).** *Fake Scene:* On a modern city street, a rebellious street performer in jeans and a rock-band T-shirt strums an electric guitar, suddenly shouting the verses in parody. *Fake Profile:* Mark, an anti-traditionalist performer who mocks classical art and heroic ideals, favoring satire and irony. *Reason:* The verses, originally meant to inspire reverence, become ironic and absurd when delivered by someone who rejects those values. The clash is extreme, generating intentional ridicule and total loss of realism.

**Discussion: Realism Example vs. Archetype Examples.**

The Yunjin case study illustrates a fundamental difference between realism evaluation and archetype evaluation. Archetype prompts, as described in Appendix D.2, are constructed by pairing abstract role categories with synthetic contextual scenes. These cover a broad space of possible situations, ranging from modern occupations (e.g., firefighters) to fictional personas (e.g., cult leaders in martial arts fantasy) or social identities (e.g., overprotective parents, laid-back friends). While such prompts provide controlled and diverse coverage, they are ultimately template-based and rely on stereotypes to elicit role-play behaviors.

In contrast, the Yunjin example is drawn from authentic performance, where the character's persona (an opera singer) and local scene (a battlefield narration) are naturally aligned through artistic delivery. The realism task does not merely test whether a model can follow a prompt, but whether it can preserve coherence between a speaker's global identity, immediate context, and expressive prosody. When Yunjin's transcript is transplanted into mismatched settings or paired with incongruent speaker profiles, the breakdown in realism becomes apparent: the same words shift from inspiring to awkward, jarring, or even absurd.

This distinction highlights the complementary strengths of the two evaluation paradigms. Archetype evaluation systematically explores a wide combinatorial space of roles and situations, but may lack the paralinguistic and contextual richness of real speech. Realism evaluation, on the other hand, anchors assessment in authentic delivery and contextual fit, providing a finer-grained lens on coherence and naturalness. Together, they ensure that role-play evaluation spans both broad stereotypical coverage and nuanced real-world plausibility.

## F.6 TTS Systems and Settings

We employed a diverse set of TTS systems to generate negative character profiles and resynthesized speech for Realism Evaluation. These included both commercial and non-commercial models, with variation in style control, conditioning prompts, and reference usage. Table 8 summarizes the systems and settings. Below, we detail the major strategies used across systems:

**Speaker Assignment.** We leveraged all publicly available speaker style information released by the corresponding websites. For each input sample, we employed `deepseek-v3` to select the best-matching speaker based on the original speech caption. In the case of `Gemini2.5TTS`, the character profile was also appended as an additional instruction to guide synthesis.

**Similar Voice Matching.** We used each platform's built-in search functionality to identify the closest available voice to the ground-truth recording. The selected voice was then used to resynthesize the utterances.

**Voice Design.** A character profile was first converted into a descriptive voice specification by prompting `deepseek-v3` (Liu et al., 2024a). This description was submitted to the TTS service to generate candidate voices. We then previewed samples via Gemini2.5Pro, selecting the voice that best matched the speaker style of the original recording before proceeding to generate speech.

**Voice Clone.** We conducted cloning using the ground-truth voice as a source, but with an intermediate representation (voice information or tag) provided to the TTS system rather than direct conditioning.

**Identical Voice Clone.** This setting differed from the above in that the ground-truth voice was directly used as conditioning input, without an intermediate tag.

**Same-Speaker Voice Clone.** Here, instead of conditioning on the specific ground-truth utterance, we used another utterance from the same speaker to guide synthesis. This tests robustness to cross-utterance generalization.

For models using voice prompts, we use an inhouse enhancement model to first process the prompt audio so as to minimize the effect of audio events and background noise.

**Inhouse Models.** We additionally employed five proprietary inhouse TTS systems, each exploring different conditioning strategies for realism-oriented speech synthesis.

- `InhouseTTS1` (Phone–Cosy2 Pipeline). The input is a phone sequence produced by a graphme-to-phoneme (G2P) front-end. A 0.4B GPT-2 model (Radford et al., 2019) predicts `CosyVoice2` S3-tokens, which are then passed through a flow-matching model to generate mel-spectrograms. A neural vocoder from `CosyVoice2` converts the mel features into audio. Both the GPT-2 and flow-matching modules are augmented with speaker embeddings for speaker control.

- `InhouseTTS2` (BPE–Cosy2 Pipeline). Instead of phones, this model takes Qwen2.5 byte-pair-encoding (BPE) tokens as input (Xu et al., 2025a). The same 0.4B GPT-2 predicts `CosyVoice2` tokens, followed by the identical flow-matching and vocoder stages as in `InhouseTTS1`. Speaker embeddings are also injected to maintain consistent speaker style.

- `InhouseTTS3` (Embedding–AudioCode Pipeline). Here, the input is Qwen2.5 BPE embeddings, which a 0.4B GPT-2 model maps directly to 4,096-dimensional audio codes. These codes are decoded via a cascaded architecture: `RepCodec` decoder (Huang et al., 2024) → content representation (with speaker embedding) → `DDIM` (denoising diffusion implicit model) module (Jeong et al., 2021) → `DAC` latent representation (Kumar et al., 2023) → `DAC` decoder. Conditioning is provided through prefix text embeddings (e.g., instructions expressing emotional states such as "angry, hateful: you betrayed me..."), along with speaker ID. The input format is `[spk_id, BPE embedding, audio codes]`.

Table 8: TTS systems and generation settings used in Realism Evaluation.

| System | Type | Prompt Type | References |
|---|---|---|---|
| OpenAI `TTS-1 HD` | Proprietary | Speaker Assignment | API Call (OpenAI, 2025c) |
| `Gemini2.5pro-preview-TTS` | Proprietary | Speaker Assignment with Instruction | API Call (Google Cloud, 2025) |
| ElevenLabs `Multilingual V2` | Proprietary
Proprietary
Proprietary | Similar Voice Matching
Voice Design
Voice Clone | API Call (ElevenLabs, 2025)
API Call (ElevenLabs, 2025)
API Call (ElevenLabs, 2025) |
| `HumeTTS` | Proprietary | Voice Design | API Call (Hume AI, 2025) |
| `OpenAudio-S1` | Proprietary
Proprietary | Identical Voice Clone
Same Speaker Voice Clone | API Call (Fish Audio, 2025)
API Call (Fish Audio, 2025) |
| MiniMax `speech-02-hd` | Proprietary
Proprietary | Voice Design
Voice Clone | API Call (Fish Audio, 2025)
API Call (Fish Audio, 2025) |
| `InhouseTTS1` | Proprietary
Proprietary | Identical Voice Clone
Same Speaker Voice Clone | -
- |
| `InhouseTTS2` | Proprietary
Proprietary | Identical Voice Clone
Same Speaker Voice Clone | -
- |
| `InhouseTTS3` | Proprietary
Proprietary | Identical Voice Clone
Same Speaker Voice Clone | -
- |
| `InhouseTTS4` | Proprietary
Proprietary | Identical Voice Clone
Same Speaker Voice Clone | -
- |
| `InhouseTTS5` | Proprietary
Proprietary | Identical Voice Clone
Same Speaker Voice Clone | -
- |
| `CosyVoice2` | Open-source
Open-source | Identical Voice Clone
Same Speaker Voice Clone | (Du et al., 2024a)
(Du et al., 2024a) |
| `Vevo1.5` | Open-source
Open-source | Identical Voice Clone
Same Speaker Voice Clone | (Zhang et al., 2025b)
(Zhang et al., 2025b) |
| `MaskGCT` | Open-source
Open-source | Identical Voice Clone
Same Speaker Voice Clone | (Wang et al., 2025d)
(Wang et al., 2025d) |
| `OpusLM-7B` | Open-source
Open-source | Identical Voice Clone
Same Speaker Voice Clone | (Tian et al., 2025)
(Tian et al., 2025) |

- `InhouseTTS4` (Instruction-Conditioned AudioCode Pipeline). This model also employs a 0.4B GPT-2 to predict 4,096-dimensional audio codes, but conditions generation on instruction embeddings. A `CLIPTextEncoder` (Radford et al., 2021) converts instruction text into embeddings, which serve as a prefix, combined with speaker ID and phone sequence. The input format is `[spk_id, instruction embedding sequence, phone sequence, audio codes]`.

- `InhouseTTS5` (Reference-Conditioned AudioCode Pipeline). Similar to `InhouseTTS4`, but instead of text-based instructions, conditioning is achieved through reference audio. A style encoder, following the NANCY++ design (Choi et al., 2023), compresses the reference audio into a fixed 32-dimensional embedding. This embedding is concatenated with the phone sequence and audio codes for generation. The input format is `[32-dim ref embedding, phone sequence, audio codes]`.

## G  ANNOTATION GUIDELINES FOR REALISM EVALUATION

This appendix contains the full human annotation instructions used for realism evaluation.

### G.1  INITIAL ASSESSMENT & REJECTION INSTRUCTIONS

- Read the **Character Profile** (long-term identity, style) and **Local Scene** (immediate context, goal/emotion) before listening.

- Focus evaluation on the *spoken performance of the target speaker only*; ignore background music, ambient noise, and non-speech events.

- If the audio's speaker is *clearly not* the intended target identity, assign a score of **1** to all Profile-dependent dimensions (e.g., Voice Identity Matching, Trait Embodiment) and proceed with other dimensions as applicable.

- If the prompt information is **contradictory** or **insufficient** to evaluate (e.g., audio too short, traits mutually incompatible), mark the prompt quality accordingly and *do not* force granular ratings that cannot be judged.

Detailed case examples and progressive gating for emotional dimensions are provided in the full instruction document.

## G.2 RUBRIC RATING INSTRUCTIONS

We adopt five dimensions, each rated from 1 (worst) to 5 (best). Emotional Expressiveness uses *progressive gating*.

- **Prosodic Dynamics** (Pitch Variation, Rhythmic Naturalness, Stress & Emphasis): naturalness of patterns and rhythms.
- **Emotional Expressiveness** (Progressive gating):
    1. Emotion Accuracy (required),
    2. Emotion Intensity Control (rate only if 2.1 $\geq$ 3),
    3. Dynamic Range (rate only if 2.2 $\geq$ 3).
- **Character Consistency** (Voice Identity Matching, Trait Embodiment): alignment with Character Profile.
- **Contextual Relevance** (Local Scene Fit, Global Story Coherence): fit to immediate scene and long-term identity.
- **Semantic Match** (spoken content vs. Local Scene goal/emotion/action).

The complete rubrics with per-score definitions and examples are reproduced below.

## G.3 COMMENTARY GUIDELINES AND CONFIDENCE RATINGS

Annotators should provide short, clear justifications:

- Use the comment box to note contradictions, missing information, vagueness, or multi-speaker issues.
- Keep comments brief and non-technical; do not quote transcripts unless necessary.
- Do not reward theatricality/technical polish unless it naturally fits the character *and* scene.

Annotators also provide a **confidence** level (1–5) reflecting familiarity with the role/scene and clarity of audible cues. Low confidence *does not* reflect poorly on annotators; it adds nuance for downstream analysis.

## G.4 FULL ANNOTATION GUIDELINE (REALISM EVALUATION)

---
**Realism Evaluation**

```
Goal.   Rate how well the given speech (audio and transcript)
aligns with the character's personality, traits, speaking
style, and intent, based on:
    • Character Profile:  the character's long-term
      personality and speaking habits.
    • Local Scene:  the current scene's context and the
      character's immediate goal or emotion.
Each subcategory is scored on a 1{5 scale.  Always read
the Character Profile and Local Scene before listening; use
both the audio and transcript (reference only) to guide your
score.
```
---

**Initial Assessment & Pre-Checks.** Before rating any speech sample, follow this checklist to ensure consistency and fairness.

CAVEATS

- **Read first:** Carefully read the *Character Profile* and *Local Scene* to understand the long-term identity and immediate goal/emotion.

- **Use both sources:** Play the audio and skim the transcript together to assess delivery quality, expression, and alignment with character intent (do not penalize transcript mismatches).

- **Focus only on human speech:** Do not factor in background music/soundtracks, environmental or ambient noise, or non-speech audio events; these may reflect the scene but should not influence scoring of the spoken performance.

- **Target speaker only:** If multiple speakers are present, evaluate only the target character's lines (consider others only as context for turn-taking).

- **Identity mismatch:** If the clip's speaker is clearly not the intended target speaker, assign a score of **1** for every rating dimension that relies on the Character Profile (e.g., Character Style Traits, Trait Embodiment, Character Consistency).

PROMPT QUALITY (PRE-CHECK)

Rate the prompt before annotation:

1. **Prompt contains contradictory information.** Conflicts make accurate annotation difficult or impossible. Examples: incompatible accent specifications; Local Scene drastically at odds with intended emotion (see *Semantic Match*); Local and global scenes inconsistent; traits described as "evolving over time" without a timeline.

2. **Prompt is insufficient to evaluate the voice sample.** Audio too short to gauge the dimensions; or the prompt lacks specific characteristics/is too vague or multi-directional to know which version of the character is present. (It is OK if audio does not match transcript exactly; if too much is missing to annotate, mark as insufficient.)

3. **Prompt is sufficient to evaluate the voice sample.** No material issues; proceed.

OVERALL SPEAKER FOCUS AND CHARACTER STYLE TRAITS

- **Checkbox 1: Multi-Speaker Influence.** Check *Yes* if other speakers were present and made it harder to isolate/judge the target character (check *Yes* if uncertain). Otherwise check *No*.

- **Question 2: Observed Character Style Traits.** Based on the Character Profile, are the defined Style Traits clearly present in the delivery?
  - Keep trait tags if a trait is expressed (positively or clearly in the opposite direction); this supports high/low embodiment scoring.
  - Delete incorrect pre-populated traits; you do not need to replace them. If you delete many traits, *Trait Embodiment* should likely be low.
  - If no traits are expressed, enter "N/A". If unsure, leave original traits unedited.

- *Example (customizable):* `"Character_style":"Playful, Charismatic, Confident, Shrewd"`.

**Rubric Rating Instructions.** Score each subcriterion on a 1–5 scale. Use the detailed rubrics below and the examples to maintain calibration across characters, tones, and emotional styles. Prioritize in-character believability; do not reward theatricality, emotional intensity, or technical polish unless they naturally fit the character *and* scene.

GUIDELINE 1. PROSODIC DYNAMICS (PATTERNS AND RHYTHMS)

Focus: pitch, rhythm, and emphasis in speech. **Guideline 1.1 Pitch Variation (check Local Scene, Character Profile, and audio).**

| Score | Description / Example |
|---|---|
| 5 | Pitch moves naturally/believably, subtly capturing emotional state & speech pattern (e.g., gentle rise when a calm-but-curious character asks a question). |
| 4 | Slight variation that fits character; somewhat restrained/inconsistent. |
| 3 | Neutral pitch/non-distracting but lacks expressive intent. |
| 2 | Forced/theatrical or mismatched pitch shifts. |
| 1 | Monotone or misaligned with scene/character. |

**Guideline 1.2 Rhythmic Naturalness (check Local Scene, Character Profile, and audio).**

| Score | Description / Example |
|---|---|
| 5 | Smooth, believable rhythm reflecting speech habit & scene pacing (e.g., nervous character in fast but coherent bursts). |
| 4 | Minor deviations but still natural. |
| 3 | Even pace without clear rhythmic personality. |
| 2 | Disjointed, overly rigid, or awkward pacing (odd pause placements). |
| 1 | Completely unnatural/robotic timing that interrupts flow. |

**Guideline 1.3 Stress and Emphasis (check Local Scene, Character Profile, and audio).**

| Score | Description / Example |
|---|---|
| 5 | Emphasis highlights key meanings/emotions in realistic ways (e.g., stress on "*can't*" in "We can't leave them behind!"). |
| 4 | Generally helpful emphasis with minor misalignment. |
| 3 | Mild/generic stress patterns (not distracting, limited impact). |
| 2 | Over-emphasis or misplaced focus (e.g., stressing function words). |
| 1 | Flat or confusing emphasis (all equal or no variation). |

GUIDELINE 2. EMOTIONAL EXPRESSIVENESS (PROGRESSIVE GATING)

Focus: alignment of expressed emotion with character style and situational demand.
**Gating:** Only rate 2.2 if $2.1 \geq 3$; only rate 2.3 if $2.2 \geq 3$.

**Guideline 2.1 Emotion Accuracy (Required; check Local Scene, Character Profile, and audio).**

| Score | Description / Example |
|---|---|
| 5 | Emotion matches personality and situation with subtlety (e.g., hesitant pause before apologizing for a reserved character). |
| 4 | Emotionally grounded with light misalignment. |
| 3 | Too neutral/safe for the scene. |
| 2 | Out of sync with character or situation (e.g., cheerful during betrayal). |
| 1 | Jarring/inappropriate (e.g., laughing/yelling without motivation). |

*If you select 1 or 2, stop here: set 2.2 and 2.3 = N/A.*

**Guideline 2.2 Emotion Intensity Control (Only if $2.1 \geq 3$; check Local Scene, Character Profile, and audio).**

| Score | Description / Example |
|---|---|
| 5 | Intensity calibrated perfectly (e.g., whispered anger in a tense standoff). |
| 4 | Mostly appropriate (slight over-/under-shoot). |
| 3 | Slightly exaggerated or dull. |
| 2 | Overacted or underplayed (e.g., screaming in a reflective moment). |
| 1 | Wildly off-scale (e.g., hysterical laughter in a death scene). |

*If you select 1 or 2, stop here: set 2.3 = N/A.*

**Guideline 2.3 Dynamic Range (Only if 2.2 ≥ 3; check Local Scene, Character Profile, and audio).**

| Score | Description / Example |
|---|---|
| 5 | Emotional shifts, if present, are natural and in-character (e.g., mentor calmly grows firmer while warning a student). |
| 4 | Appropriate shifts present but mild/slow to emerge. |
| 3 | Mostly steady; slight modulation; no strong arc. |
| 2 | Flat/monotone or unnaturally varied with unjustified jumps. |
| 1 | Significant out-of-character fluctuations without triggers. |

GUIDELINE 3. CHARACTER CONSISTENCY

Focus: how well the voice and language align with the character's core identity.

**Guideline 3.1 Voice Identity Matching (Character Profile, and audio).**

| Score | Description / Example |
|---|---|
| 5 | Voice clearly matches the expected identity (e.g., youthful, curious tone for energetic sidekick). |
| 4 | Acceptable match with minor mismatches. |
| 3 | Neutral/not character-specific. |
| 2 | Somewhat inconsistent (e.g., deep voice for high-pitched character). |
| 1 | Clearly mismatched identity (e.g., gritty adult voice for bubbly child). |

**Guideline 3.2 Trait Embodiment (Character Profile, and audio).**

| Score | Description / Example |
|---|---|
| 5 | Defining traits subtly but clearly conveyed (e.g., insecure character hesitates/lowers voice). |
| 4 | Most traits conveyed; minor inconsistencies. |
| 3 | Traits weak but not disruptive. |
| 2 | Missing or conflicting traits (e.g., arrogant tone for a humble character). |
| 1 | No identifiable traits or completely off. |

GUIDELINE 4. CONTEXTUAL RELEVANCE

Focus: appropriateness in the current scene and long-term coherence.

**Guideline 4.1 Local Scene Fit (check Local Scene, and audio).**

| Score | Description / Example |
|---|---|
| 5 | Matches the scene's emotional/narrative context (e.g., urgent whisper during stealth). |
| 4 | Mostly coherent with minor mismatch (e.g., calm tone while others panic). |
| 3 | Neutral: neither reinforces nor harms the scene. |
| 2 | Confusing or slightly inappropriate tone (e.g., laughing during a tense standoff). |
| 1 | Directly contradicts scene intent (e.g., casual joke while a character is dying). |

**Guideline 4.2 Global Story (Character Profile) Coherence (check Character Profile, and audio).**

| Score | Description / Example |
|---|---|
| 5 | Perfectly consistent with background and arc (e.g., wise mentor advice aligned with past scenes). |
| 4 | Slightly off-style but plausible (e.g., cheerful tone for usually sarcastic character). |
| 3 | Generic/filler delivery (*could be anyone*). |
| 2 | Contradicts growth or habits (e.g., arrogance after humility arc). |
| 1 | Breaks established identity (e.g., cold anger from consistently empathetic healer). |

GUIDELINE 5. SEMANTIC MATCH

Focus: how well the spoken content semantically aligns with the Local Scene's stated goal, emotion, or action.

**5.1 Semantic Matchness (Audio, Local Scene).**

| Score | Description / Example |
|---|---|
| 5 | Content perfectly advances the scene goal/emotion (e.g., rescue: "Secure the exit; I'll get the hostages."). |
| 4 | Minor nuance mismatch but still scene-appropriate. |
| 3 | Neutral filler that neither helps nor contradicts (plausible bridge). |
| 2 | Partially conflicts or confuses context; requires *stretch* explanation. |
| 1 | Directly contradicts/derails scene intent; no reasonable bridge. |

CONFIDENCE

Provide a confidence rating (1–5) reflecting your certainty that you:

- Understood the character and scene clearly,

- Heard sufficient emotional cues in the audio, and

- Interpreted speech alignment with intended personality and context.

**Guidance:** Rate low confidence for noisy/ambiguous audio, vague/contradictory traits, multiple dominant speakers, or hard-to-distinguish emotion/tone.

COMMENTS (OPTIONAL)

Use the comment box to note contradictions, missing information, vague descriptions, or multi-speaker issues that affected your evaluation. Keep comments brief, clear, and non-technical; no transcript quotes are required.

# H  ANNOTATION METHODOLOGY: COMPARISON TO CONTEMPORARY WORK

## H.1  SCOPE AND LABEL SOURCE

Figure 2 contrasts how recent resources obtain labels and what they optimize for.

## H.2  OUR ANNOTATION DESIGN AND WHY IT MATTERS

**Human-grounded, bilingual labels.** We annotate in English *and* Mandarin with trained raters, capturing cross-lingual prosody and style, while recording rater confidence to qualify downstream usage.

**Dual settings: Archetype vs. Realism.** We deliberately separate stereotype-driven *Archetype* from bottom-up *Realism* tasks. This avoids averaging away nuance: archetype labels reflect schema-level fit; realism labels prioritize fine-grained, in-situ delivery from real-world or carefully constructed contrastive material.

Figure 2: Annotation pipeline comparison across three representative frameworks.

|  | SpeechRole | Audio-Aware LLMs as Judges | InstructTTSEval |
|---|---|---|---|
| **Data basis** | LLM-generated dialogues (∼112k) for 98 roles; speech via collection/synthesis; limited manual verification | Model outputs evaluated by audio-aware LLMs (ALLM-as-judge) on voice-style IF and role-play | Expressive clips mined from movies/TV; reverse-generated style instructions (EN/ZH); Gemini-as-judge |
| **Label source** | Mixture: automatic pipelines + manual checks; benchmark dims: interaction/-expressiveness/role fidelity | Automatic labels from ALLMs; no human rubric for each clip during evaluation | Automatic labels from Gemini; human annotations used for agreement calibration |
| **Primary goal** | Large-scale SRPA dataset & benchmark across paradigms (cascaded/E2E) | Feasibility of ALLM-as-judge for speaking style/role-play | Automatic benchmarking of instruction-following TTS across APS/DSD/RP |
| **Key limitation (for DRAME's aims)** | Synthetic bias from LLM/TTS; conflates generator priors with evaluation targets | Judge sensitivity to prompts; opaque criteria; limited human verifiability | Judge-driven scores; limited per-dimension human rationale; coarse control over rater bias |

**Progressive gating for emotion.** Our *Emotional Expressiveness* uses gated sub-scores (accuracy → intensity control → dynamic range), preventing spurious high scores when base emotion is misidentified. This reduces label noise and improves SEM alignment stability.

**Contrastive negatives and semantic controls.** Realism includes profile/scene mismatches and re-synthesized counterfactuals to force evaluators (human/SEM) to use paralinguistic cues, not only transcript semantics.

**Auditability and SEM alignment.** Every score is traceable to a rubric with short rater rationales and confidence. We then tune SEMs to these labels (rather than opaque judge outputs), enabling calibration studies and error analyses that would not be possible with black-box LLM judges.

### H.3 WHY SPEECH DRAME'S LABELS ARE PREFERABLE FOR ROLE-PLAY ASSESSMENT

1. **Sensitivity to delivery.** Human raters, bilingual scope, and gated rubrics capture intonation, pacing, and subtle emotional control that judge-only pipelines often miss or compress into single numbers (see Appdendix G).

2. **Separation of concerns.** By isolating *Archetype* vs. *Realism* we (i) retain scalability where stereotypes are enough, and (ii) do fine-grained realism without stereotype leakage.

3. **Reduced synthetic bias.** Unlike pipelines built from LLM-generated dialogues or TTS-only references, we anchor annotations to real-world/contrastive human recordings and explicitly stress-test mismatches (Appdendix F.4, Appdendix F.6).

4. **Transparent evaluation substrate.** Ours produces per-dimension human rationales and confidence; judge-only pipelines (ALLM/Gemini) provide scores with limited per-sample interpretability (Chiang et al., 2025; Huang et al., 2025b).

5. **Better for SEM training.** Rubric-aligned labels let us *train* and *audit* SEMs, measure human alignment, and analyze error modes; judge outputs are harder to trust long-term as they change with model versions and prompts.

## I  ANNOTATION ETHICS

All human annotation in this work was conducted in accordance with standard ethical practices for research involving human annotators. Annotators were recruited via public hiring for Mandarin and internal lab hiring for English. Prior to participation, all annotators were provided with a clear description of the task, the type of data they would encounter, and their rights as participants, including the ability to opt out at any time without penalty.

Annotators gave informed consent before starting the tasks. They were compensated at a rate from $20 to $30 per hour, which corresponds to at least the local minimum wage and was applied uniformly regardless of whether their contributions were later filtered for quality.

The data presented to annotators consisted of synthetic speech samples, anonymized human speech, or recordings from public corpora, and contained no personally identifiable or sensitive information. To mitigate potential risks, annotators were allowed to skip any item that they found uncomfortable or inappropriate.

To ensure annotation reliability, annotators were provided with written guidelines, examples, and trial tasks before beginning. Quality control procedures included inter-annotator agreement checks, and comprehensive screening tests. Each speech sample is evaluated by at least 3 annotators. This study did not involve vulnerable populations and posed minimal risk to participants.

## J  DETAILED DATA ANALYSIS (ARCHETYPE EVALUATION)

The archetype evaluation provides a detailed view of how speech foundation models (SFMs) perform when tasked with role-play scenarios grounded in stereotypical characters and situations. Human annotations cover four dimensions, *Appropriateness*, *Human Likeness*, *Audio Quality*, and *Content Pass*, and the results reveal important trends, as well as challenges in interpreting scores across languages.

**Annotation Consistency and Cross-Language Comparisons.**    A critical observation is that annotations in Mandarin and English cannot be directly compared. The annotation groups for the two languages differed in their linguistic backgrounds and cultural interpretations, which leads to systematic differences in scoring. Our bilingual pilot further confirmed that annotators often interpret role-play appropriateness and expressiveness differently depending on their cultural frame of reference. For example, Mandarin raters tend to reward delivery styles that align with culturally expected prosody and politeness strategies, while English raters prioritize spontaneity and naturalness. These discrepancies parallel the findings of the singing voice conversion challenge (SVCC2023) from Huang et al. (2023), where native and non-native listeners provided significantly different ratings for the same samples, particularly for naturalness and speaker similarity. Consequently, while absolute scores differ substantially across Mandarin and English, only within-language comparisons are meaningful. Cross-lingual ranking is not valid given the different annotation bases.

**Mandarin Results.**    Within Mandarin from Table 9, the relative rankings of models are clear. `Doubao` stands out as the strongest performer across all four dimensions, reaching near-ceiling levels in both *Audio Quality* (4.37) and *Content Pass* (0.98). It also leads in role-related dimensions such as *Appropriateness* and *Human Likeness*, indicating balanced competence across linguistic accuracy, delivery, and expressiveness. `G-CosyVoice2` and `Q-CosyVoice2` follow closely, especially in audio quality, though they lag behind in role appropriateness. `KimiAudio` and `GLM4-Voice` achieve mid-range results, reflecting higher variance across samples, while `GPT-4o-realtime` shows competitive content understanding but lower naturalness and appropriateness. `Qwen2.5Omni` consistently ranks the lowest, with large variance, underscoring instability and weaker alignment with archetypal role expectations.

Table 9: Overall human annotation results on Mandarin and English archetype evaluation dimensions (mean $\pm$ std). Best per column in **bold**.

| Model (Mandarin) | Appropriateness | Human Likeness | Audio Quality | Content Pass |
|---|---|---|---|---|
| Doubao | **3.8236 $\pm$ 0.6658** | **3.6572 $\pm$ 0.6427** | **4.3743 $\pm$ 0.6991** | **0.9848 $\pm$ 0.1224** |
| G-CosyVoice2 | 3.4243 $\pm$ 0.7389 | 3.3156 $\pm$ 0.7215 | 4.0406 $\pm$ 0.7298 | 0.9725 $\pm$ 0.1637 |
| Q-CosyVoice2 | 3.4228 $\pm$ 0.7788 | 3.2750 $\pm$ 0.7335 | 4.0236 $\pm$ 0.7661 | 0.9674 $\pm$ 0.1776 |
| KimiAudio | 3.3866 $\pm$ 1.1086 | 3.3391 $\pm$ 1.0591 | 3.9525 $\pm$ 1.2801 | 0.8623 $\pm$ 0.3446 |
| GLM4-Voice | 3.2022 $\pm$ 0.9597 | 3.0275 $\pm$ 0.8687 | 3.9246 $\pm$ 1.1105 | 0.8949 $\pm$ 0.3067 |
| GPT-4o-realtime | 3.0228 $\pm$ 0.8581 | 2.6536 $\pm$ 0.7151 | 3.7297 $\pm$ 1.0418 | 0.9098 $\pm$ 0.2865 |
| Qwen2.5Omni | 2.4942 $\pm$ 1.4242 | 2.3830 $\pm$ 1.3201 | 2.8685 $\pm$ 1.7456 | 0.5482 $\pm$ 0.4978 |
| **Model (English)** | | | | |
| ChatGPT-4o | **3.4638 $\pm$ 0.9671** | **3.4873 $\pm$ 0.7857** | 1.9620 $\pm$ 0.6885 | **0.9662 $\pm$ 0.1808** |
| Doubao | 2.3798 $\pm$ 1.0573 | 2.2355 $\pm$ 0.8918 | 2.9553 $\pm$ 1.1589 | 0.7995 $\pm$ 0.4005 |
| GPT-4o-realtime | 2.2736 $\pm$ 0.9606 | 2.0906 $\pm$ 0.8124 | 2.5127 $\pm$ 0.9702 | 0.8521 $\pm$ 0.3552 |
| GLM4-Voice | 2.2355 $\pm$ 1.0354 | 2.0779 $\pm$ 0.9095 | 2.5694 $\pm$ 1.1366 | 0.7482 $\pm$ 0.4342 |
| G-CosyVoice2 | 2.2168 $\pm$ 0.9916 | 2.0109 $\pm$ 0.8585 | **3.2530 $\pm$ 0.9637** | 0.9293 $\pm$ 0.2563 |
| Q-CosyVoice2 | 2.1673 $\pm$ 0.9642 | 1.8068 $\pm$ 0.8011 | 3.2476 $\pm$ 0.9113 | 0.9614 $\pm$ 0.1928 |
| Qwen2.5Omni | 2.0079 $\pm$ 1.0420 | 1.9408 $\pm$ 0.9489 | 2.4161 $\pm$ 1.2583 | 0.6159 $\pm$ 0.4865 |
| KimiAudio | 1.5531 $\pm$ 0.8716 | 1.7579 $\pm$ 0.9386 | 2.0097 $\pm$ 1.1788 | 0.4861 $\pm$ 0.5000 |

Table 10: Domain-level human annotation results for Appropriateness dimension of Mandarin and English archetype evaluation (mean $\pm$ std). We report results of two languages with inter-model variance separately.

| Domain | Mandarin | | English | |
|---|---|---|---|---|
| | Score (mean $\pm$ std) | Inter-model Var. | Score (mean $\pm$ std) | Inter-model Var. |
| NamedCharacters | 3.0343 $\pm$ 1.1087 | 1.1087 | 1.9896 $\pm$ 1.0409 | 1.0409 |
| FantasyOccupation | 3.1629 $\pm$ 0.9661 | 0.9661 | 1.9444 $\pm$ 0.9809 | 0.9809 |
| Traits | 3.1670 $\pm$ 1.2109 | 1.2109 | 2.1518 $\pm$ 1.0572 | 1.0572 |
| Events | 3.2535 $\pm$ 1.1523 | 1.1523 | 2.2608 $\pm$ 1.1280 | 1.1280 |
| DailyOccupation | 3.2760 $\pm$ 0.8652 | 0.8652 | 2.5129 $\pm$ 1.1781 | 1.1781 |
| RelativeRoles | 3.2762 $\pm$ 1.0255 | 1.0255 | 2.3594 $\pm$ 1.1298 | 1.1298 |
| SocialIdentity | 3.3263 $\pm$ 0.9802 | 0.9802 | 2.3256 $\pm$ 1.0912 | 1.0912 |

**English Results.** The picture shifts in English, elaborated in Table 9. Here, `ChatGPT-4o` emerges as the most effective model by a wide margin, achieving top scores in both *Appropriateness* (3.46) and *Human Likeness* (3.49), as well as high content pass rates (0.97). This reflects its strong grounding in English discourse norms and role adherence. In contrast, `CosyVoice2` variants perform well in *Audio Quality* but are weaker in role alignment, highlighting a trade-off between acoustic fidelity and role-play embodiment. Other models such as `Doubao`, `GLM4-Voice`, and `GPT-4o-realtime` achieve moderate results, while `KimiAudio` and `Qwen2.5Omni` trail far behind, showing limited robustness in English scenarios. Taken together, the English results reveal both a sharper separation across models and a wider variance in appropriateness scores compared to Mandarin.

**Domain-Level Trends.** Table 10 and Figures 3–4 further highlight differences across domains. In Mandarin, performance is relatively consistent across categories, with means clustering around 3.1–3.3. The highest scores appear in `SocialIdentity` and `DailyOccupation`, suggesting that models handle culturally grounded archetypes more reliably. In English, however, domain variation is much greater. Categories such as `FantasyOccupation` and `NamedCharacters` receive notably lower scores (means below 2.0), reflecting the difficulty of producing expressive, role-appropriate speech in imaginative or culturally specific contexts. Occupational roles remain relatively easier for models in both languages, likely because these scenarios are closer to training data distributions.

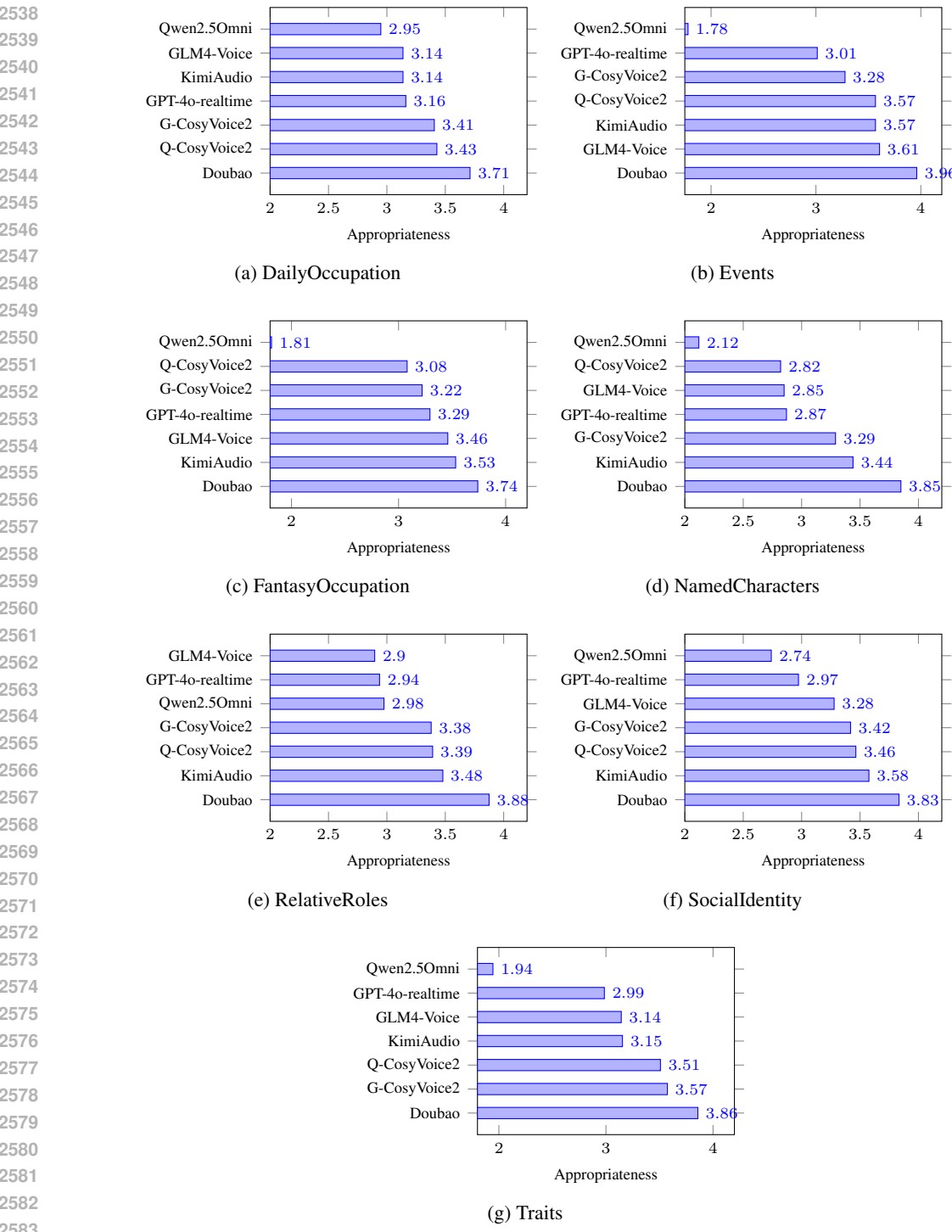

Figure 3: Model rankings within domains for Appropriateness dimension of **Mandarin** archetype evaluation (higher is better). Each bar shows the mean score for that domain.

**Interpretation and Broader Implications.** Although cross-lingual comparisons are invalid, this does not diminish the utility of archetype evaluation for role-play benchmarking. In fact, the observed inconsistencies reinforce a central point: successful role-play requires models to adapt to the cultural and linguistic norms of the target language community. A system that performs well in Mandarin but poorly in English cannot be considered universally capable, since end users will

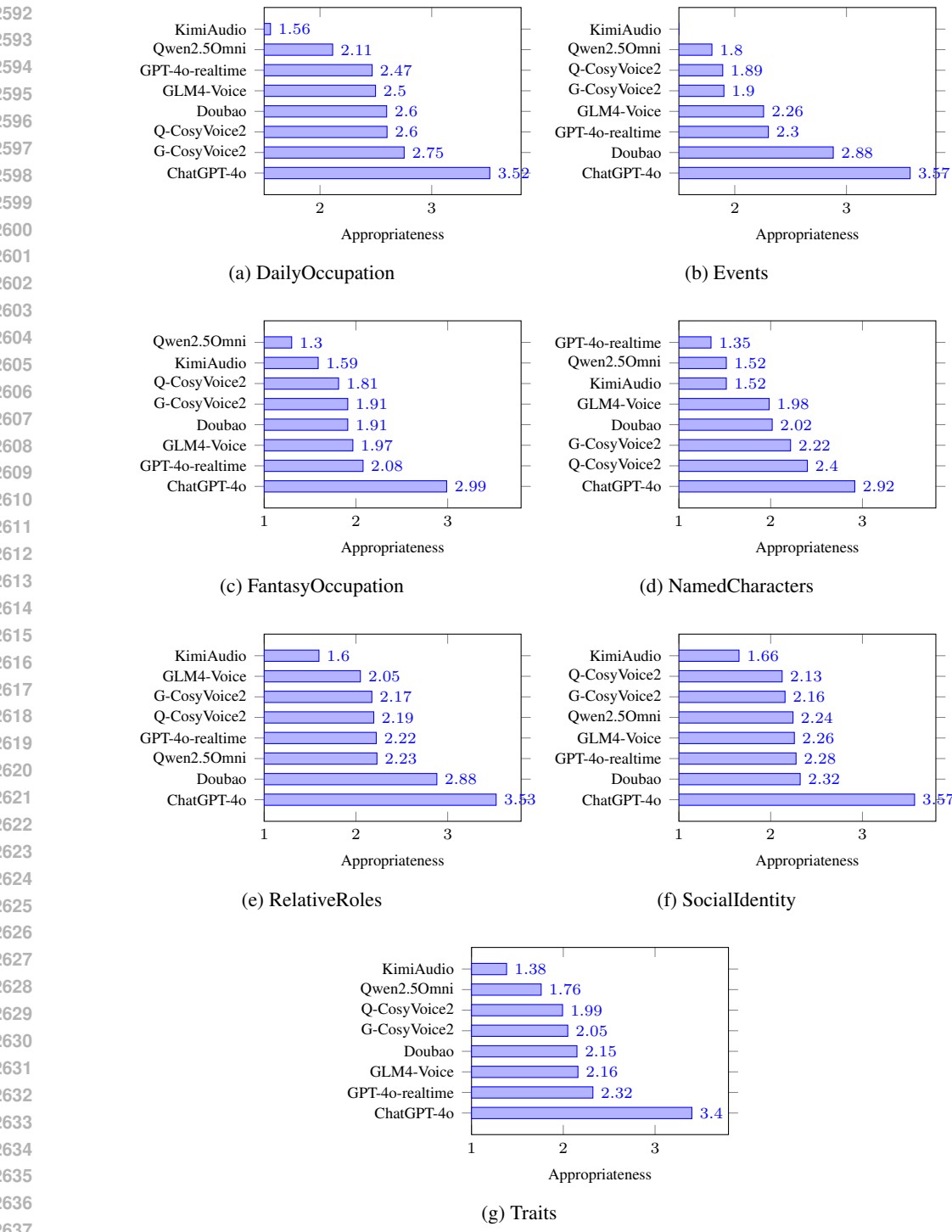

Figure 4: Model rankings within domains for Appropriateness dimension of **English** archetype evaluation (higher is better). Each bar shows the mean score for that domain.

evaluate appropriateness and naturalness according to their native expectations. Thus, archetype evaluation highlights not only model-specific strengths and weaknesses but also the need for culturally grounded evaluation pipelines.

Another notable pattern is the trade-off between acoustic quality and role embodiment. CosyVoice2 variants achieve strong audio quality but weaker human likeness and appropriateness, whereas `ChatGPT-4o` prioritizes role coherence at the expense of audio fidelity. This suggests that future systems must better integrate high-quality synthesis with nuanced role adherence. Similarly, the consistently lower scores for imaginative and identity-driven roles underscore the limitations of current SFMs in capturing subtle sociocultural expectations and narrative context.

**Summary.** In sum, archetype evaluation reveals three major findings: (1) Mandarin and English annotations cannot be directly compared, due to linguistic and cultural annotation inconsistencies, consistent with prior evidence from SVCC 2023. (2) Within each language, clear model rankings emerge, `Doubao` dominates in Mandarin, while `ChatGPT-4o` excels in English. (3) Domain sensitivity remains a challenge, particularly for imaginative or culturally rich roles, where scores are substantially lower and inter-model variance is higher.

## K  DETAILED DATA ANALYSIS (REALISM EVALUATION)

**Overview.** Although realism evaluation targets nuanced paralinguistic behaviors, human annotators achieve *consistently high agreement* once the rubric disentangles the aspects to be judged. As shown in the agreement summary in Fig. 5, most dimensions fall in the 0.86–0.88 range, with the lowest agreement on *Traits Embodiment* ($\approx 0.83$) and the highest on *Local Scene Fit / Semantic Match* ($\approx 0.88$). Emotion-related, progressive metrics (Intensity and Transition) are slightly lower ($\approx 0.85$), reflecting the added difficulty of judging graded expressivity.

**Agreement definition.** Let $R$ denote the possible score range (e.g., $R = 4$ for a 1–5 scale). For a sample $i$ with annotator scores $\{s_{ij}\}_j$, we compute the sample standard deviation $\sigma_i$ and define the per-sample agreement

$$a_i = \max\big(0, \min(1, 1 - \sigma_i/R)\big).$$

We then report the overall agreement $A = 1 - \bar{\sigma}/R$, where $\bar{\sigma}$ is the mean of $\{\sigma_i\}$ over samples with at least two annotations. We also clamp $A$ to $[0, 1]$ and ignore single-annotator cases. A minimal, reproducible implementation is:

Listing 1: Agreement computation (concise).

```python
def compute_agreement(values_by_sample, max_possible_range):
    stds, per_sample = [], {}
    for i, scores in enumerate(values_by_sample):
        x = [float(s) for s in scores if s is not None]
        if len(x) < 2: continue
        x = np.asarray(x, dtype=float)
        s = x.std(ddof=1)
        a = 1 - (s / max_possible_range) if max_possible_range > 0 else 1
        a = max(0, min(1, a))
        per_sample[f"sample_{i}"] = {
            "scores": x.tolist(), "mean": round(x.mean(), 3),
            "std": round(s, 3), "range": round(x.max()-x.min(), 3),
            "agreement_score": round(a, 3), "annotator_count": len(x)
        }
        stds.append(s)
    mean_std = np.mean(stds) if stds else np.nan
    A = 1 - (mean_std / max_possible_range) if stds else np.nan
    if not np.isnan(A): A = max(0, min(1, A))
    return {"mean_intra_sample_std": mean_std,
            "sample_agreements": per_sample,
            "overall_agreement": A}
```

**Distributional characteristics.** Fig. 6–7 visualize the score distributions per dimension. We highlight three patterns: (i) **Progressive/gated emotion metrics.** For *Emotion Intensity* and *Emotion Transition*, we assign 1 when the scene is marked *N/A* (i.e., emotion accuracy is insufficient to meaningfully judge intensity/transition). This produces a visible mass at 1, a sparse bin at 2, and slightly lower overall agreement; downstream analyses conditioning on adequate *Emotion Accuracy*

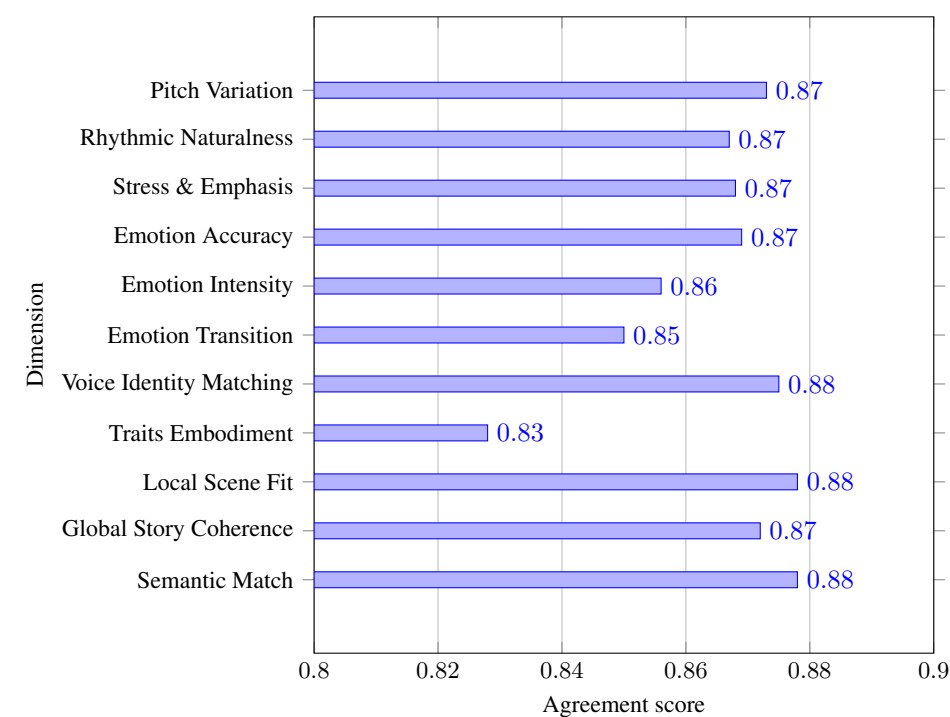

Figure 5: Agreement scores across evaluation dimensions. Higher values indicate stronger annotator consistency.

(e.g., mean $\geq 3$) recover more bell-shaped distributions. (ii) **Traits Embodiment.** Scores tend to avoid the middle ($\sim 3$), concentrating on low values when traits clearly mismatch and high values when traits are strongly realized. This yields the lowest inter-rater agreement among dimensions, reflecting the inherently compositional and role-specific nature of trait judgments. (iii) **Other dimensions.** *Pitch Variation*, *Rhythmic Naturalness*, *Stress & Emphasis*, *Voice Identity Matching*, *Local Scene Fit*, *Global Story Coherence*, and *Semantic Match* are approximately unimodal with modes at 3–4, consistent with moderate–strong realization in the dataset and with the high agreement reported.

**Professional vs. Amateur voice actors.** The paired histograms in Figs. 8–18 compare professional (left) and amateur (right) recordings:

- **Prosody control (Pitch, Rhythm, Stress).** Professional speech is shifted toward higher bins with tighter spread, indicating steadier control and fewer low-end tails.

- **Emotion rendering (Accuracy, Intensity, Transition).** Professionals show higher centers and reduced mass at 1 (fewer gated N/As), suggesting more consistent emotional grounding and smoother dynamics; amateurs exhibit wider variance and more mass at the extremes.

- **Identity and traits (Voice Identity, Traits Embodiment).** Professionals present clearer peaks in upper bins, while amateurs are broader or bimodal, reflecting inconsistent character anchoring.

- **Scene and semantics (Local Scene Fit, Global Coherence, Semantic Match).** Differences are present but smaller; both groups cluster in the mid–high range, with professionals exhibiting slightly narrower distributions.

Overall, professionals outperform amateurs on expressivity and control, while narrative/semantic alignment shows a milder gap.

**Implications for modeling.** First, the consistently high agreement confirms that the rubric isolates perceptually coherent factors, so learned evaluators can target them reliably. Second, the gated

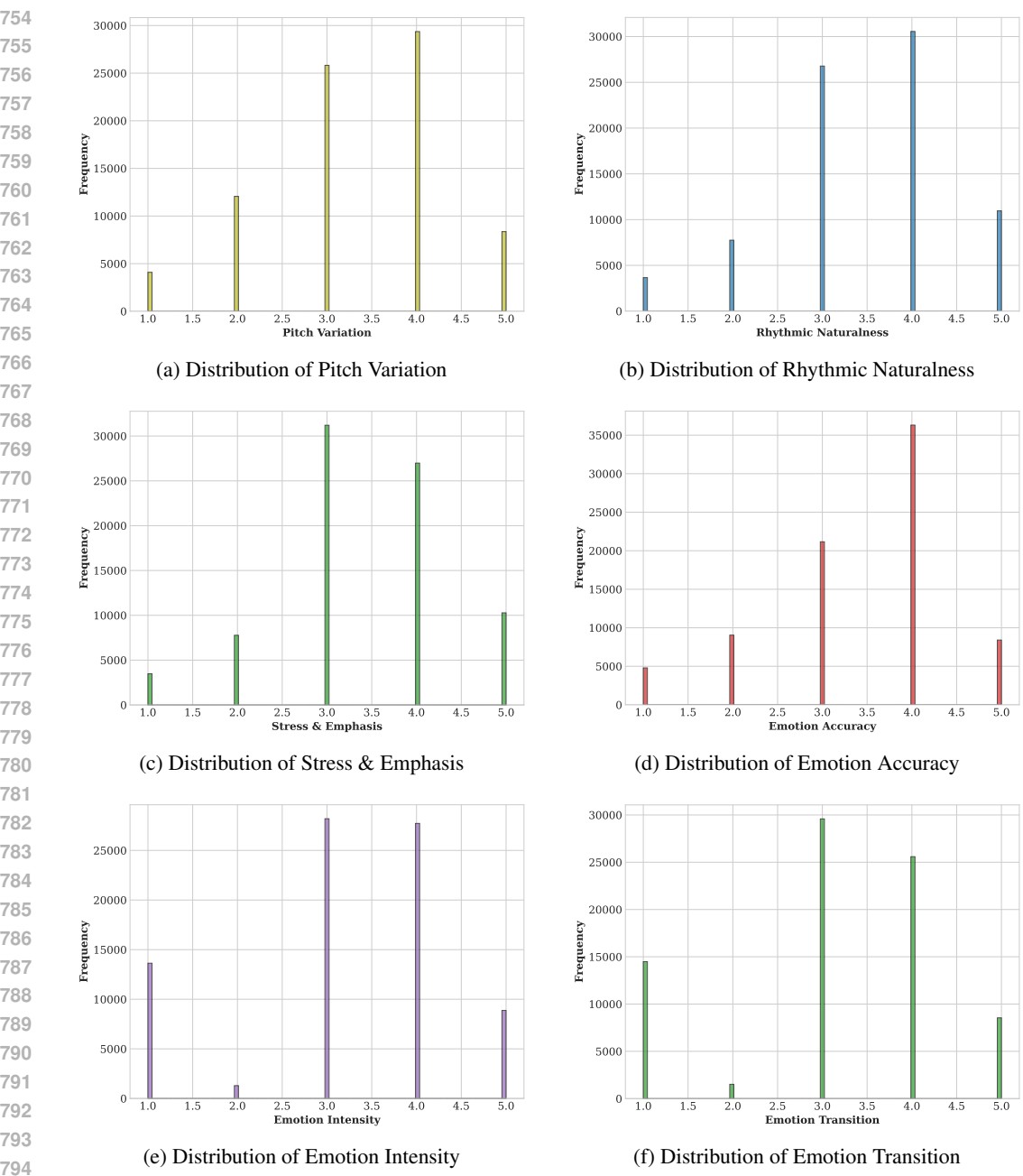

(a) Distribution of Pitch Variation

(b) Distribution of Rhythmic Naturalness

(c) Distribution of Stress & Emphasis

(d) Distribution of Emotion Accuracy

(e) Distribution of Emotion Intensity

(f) Distribution of Emotion Transition

Figure 6: Distributions of evaluation dimensions in Realism Evaluation (1).

behavior of *Intensity/Transition* motivates a *hierarchical* evaluator: verify *Emotion Accuracy* before scoring graded expressivity, or model the joint structure with conditional heads. Third, *Traits Embodiment* remains challenging; models may benefit from explicit character-trait priors, long-term context aggregation, or contrastive supervision between matched and mismatched traits. Finally, prosody-focused objectives (e.g., pitch/rhythm/stress control) are likely to yield the largest quality gains for amateur-style speech.

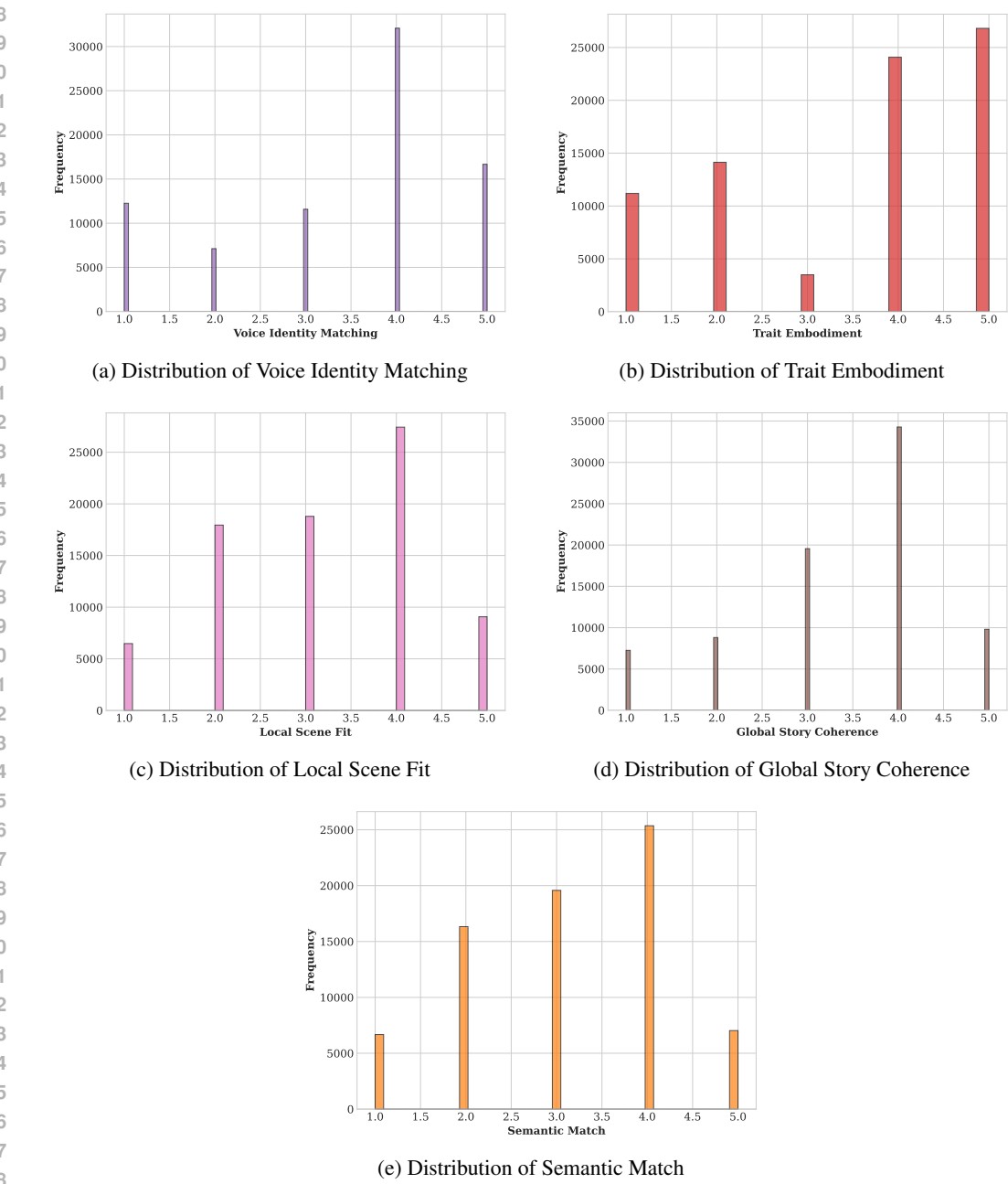

(a) Distribution of Voice Identity Matching

(b) Distribution of Trait Embodiment

(c) Distribution of Local Scene Fit

(d) Distribution of Global Story Coherence

(e) Distribution of Semantic Match

Figure 7: Distributions of evaluation dimensions in Realism Evaluation (2).

## L  EVALUATION MODEL EXPERIMENTS

### L.1  ZEROSHOT/FEWSHOT SETUP FOR EVALUATION MODELS

For both zeroshot and fewshot experiments, we adopt a streamlined version of the annotation guidelines to ensure consistency with the human annotation process. The exact rubric prompts for archetype and realism evaluation used are provided below:

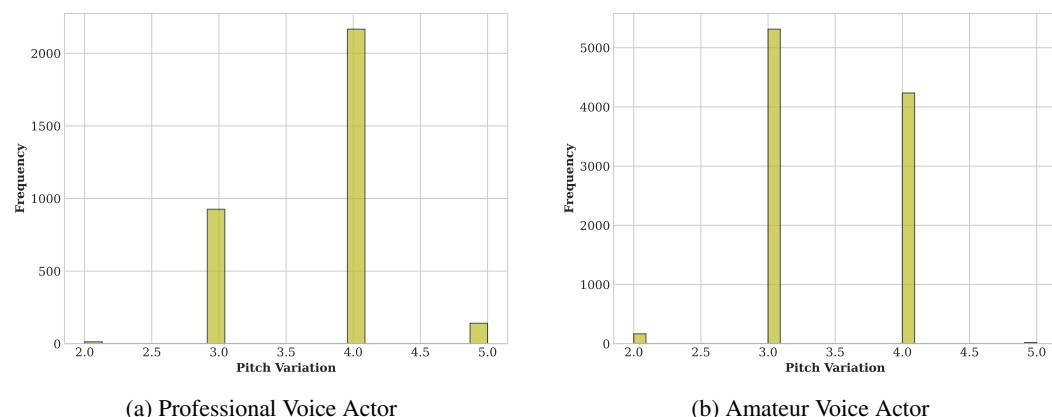

(a) Professional Voice Actor                    (b) Amateur Voice Actor

Figure 8: Distribution of Pitch Variation in Realism Evaluation: Professional vs. Amateur.

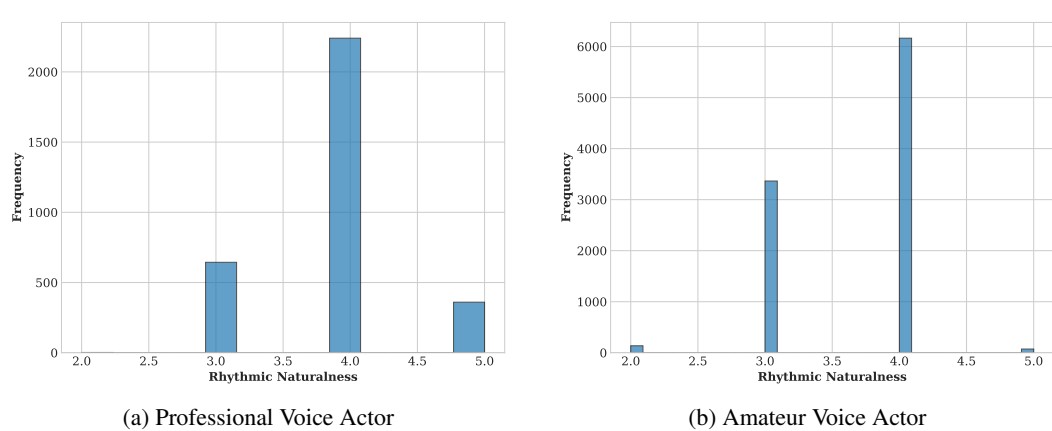

(a) Professional Voice Actor                    (b) Amateur Voice Actor

Figure 9: Distribution of Rhythmic Naturalness in Realism Evaluation: Professional vs. Amateur.

---

**Rubrics Prompts (Archetype Evaluation)**

```
Follow the rubric EXACTLY and return ONLY a minified JSON
object (valid for json.loads) | no extra text.
Set content_pass to false (i.e., reject) if ANY apply:
- Clip ~1{2s (too short to judge).
- Obvious breakdown/loop/restart.
- Any multilingual word/phrase (except extremely common
one-offs like ``croissant'').
- Irrelevant to prompt's role/scene.
- Instruction failure:  breaks persona/scene, 4th-wall
narration, multi-party narration, long placeholder
scaffolding.
Otherwise:  PASS.

WHAT TO DO
1) Apply the rubrics below strictly.  If torn between two
scores, choose the LOWER one.
2) Return ONLY a minified JSON object (valid for json.loads).
Integers 1{5.  No extra text.
3) If not rejected, judge performance (HOW it's said), not
```

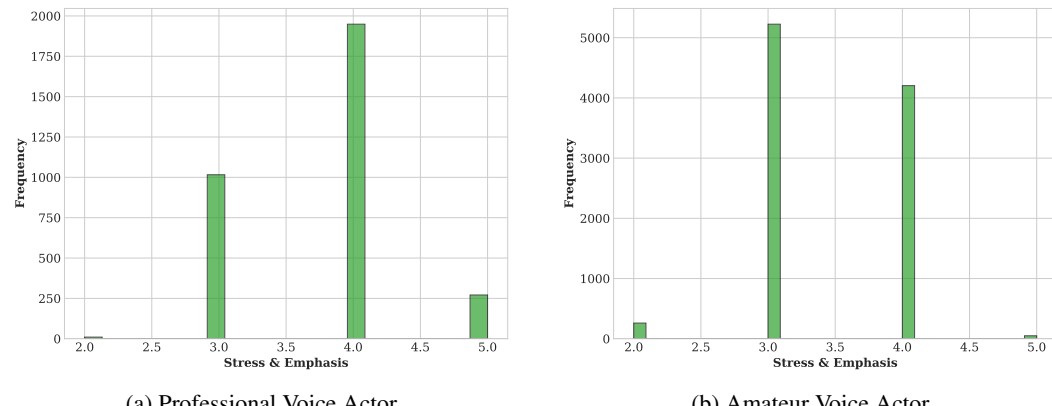

(a) Professional Voice Actor            (b) Amateur Voice Actor

Figure 10: Distribution of Stress & Emphasis in Realism Evaluation: Professional vs. Amateur.

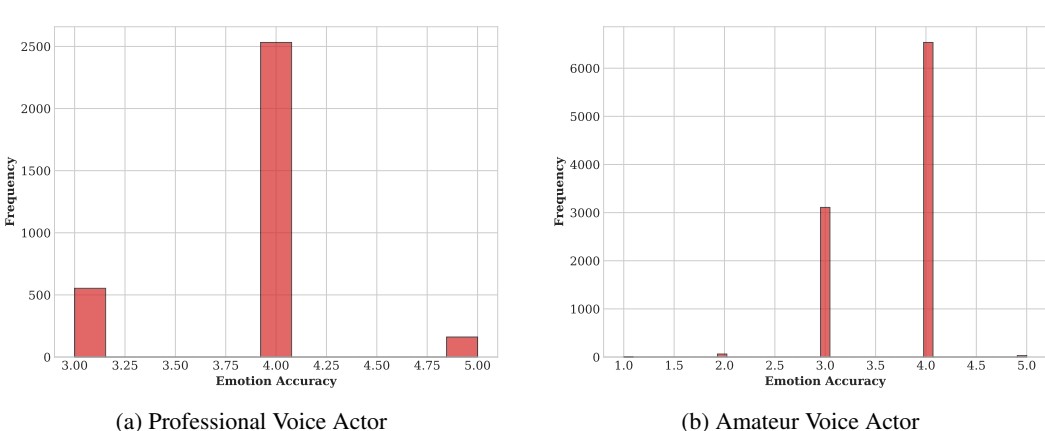

(a) Professional Voice Actor            (b) Amateur Voice Actor

Figure 11: Distribution of Emotion Accuracy in Realism Evaluation: Professional vs. Amateur.

```
wording.
4) If rejected, set all scores to 1.

0) Content-Pass
If rejected, set to false, otherwise set to true.

1) AudioQuality (Context-Free)
Focus ONLY on digital/synthetic artifacts (noise, hiss,
metallic timbre, clipping, glitchy transitions, pitch
collapse, static).
After you record this score, PRETEND these issues do not
exist for later dimensions.
Scale:
- 5 (Excellent):  Clean/clear; no noticeable artifacts.
- 4 (Good):  Minor imperfections (slight hiss/sharpness)
under close listen.
- 3 (Fair):  Noticeable artifacts (buzz/echo/synthetic
texture) that impact clarity.
- 2 (Poor):  Frequent glitches/clipping/metallic
resonance/harsh transitions.
```

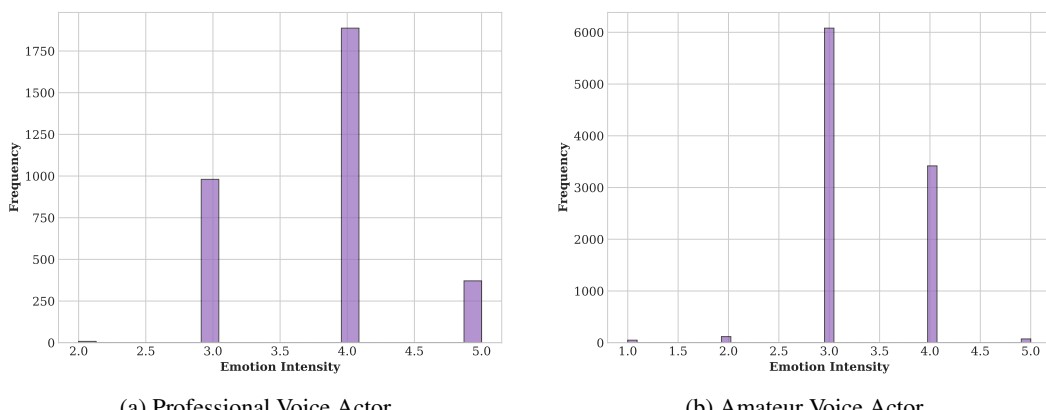

(a) Professional Voice Actor         (b) Amateur Voice Actor

Figure 12: Distribution of Emotion Intensity in Realism Evaluation: Professional vs. Amateur.

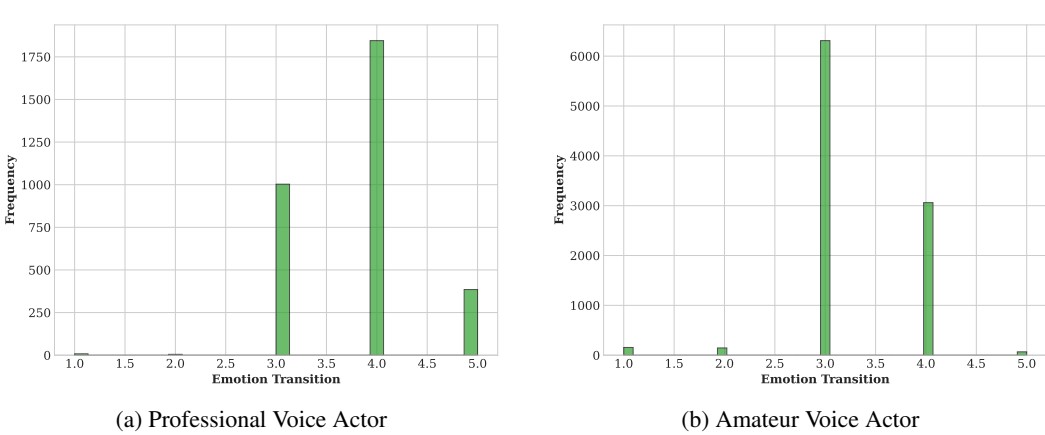

(a) Professional Voice Actor         (b) Amateur Voice Actor

Figure 13: Distribution of Emotion Transition in Realism Evaluation: Professional vs. Amateur.

```
- 1 (Very Poor):  Dominated by artifacts
(static/crackling/synthesis failure/severe clipping or pitch
collapse).

2) HumanLikeness (Context-Free)
How human-like is the delivery?  Ignore AudioQuality issues
(already captured) and ignore contextual fit.
Scale:
- 5 (Definitely Human):  Indistinguishable; rich intonation,
natural real-time pacing, consistent identity; organic
breaths/pauses.
- 4 (Most Likely Human):  Generally convincing; minor
stiffness or overly even timing/pauses.
- 3 (Could be Human or AI): Mixed traits; some mechanical
pauses/inflection, other parts human-like.
- 2 (Mostly Likely AI): Clearly artificial; choppy rhythm,
flat/illogical inflections, awkward or forced breaths.
- 1 (Definitely AI): Rigid read-aloud monotony; no natural
rhythm/pauses; entirely machine-like.
```

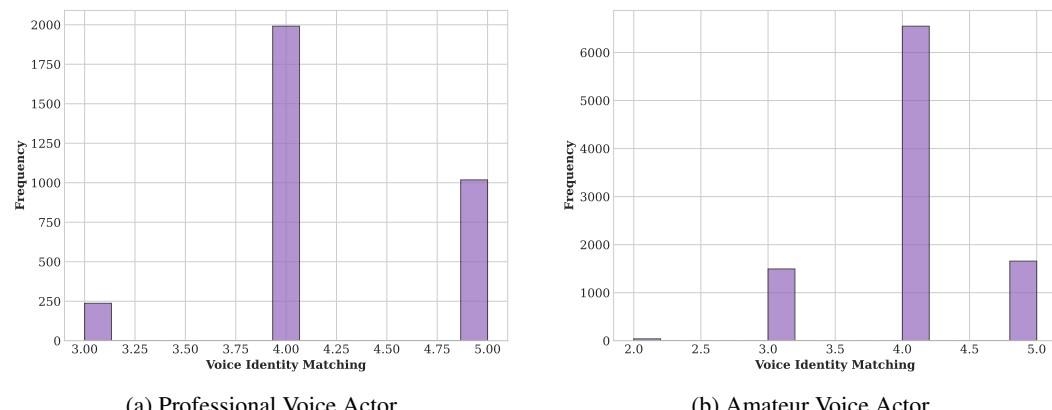

(a) Professional Voice Actor       (b) Amateur Voice Actor

Figure 14: Distribution of Voice Identity Matching in Realism Evaluation: Professional vs. Amateur.

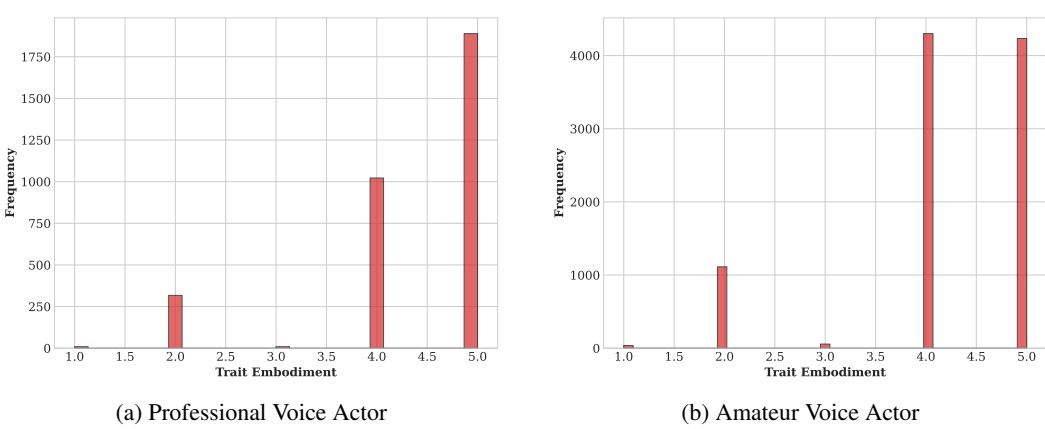

(a) Professional Voice Actor       (b) Amateur Voice Actor

Figure 15: Distribution of Traits Embodiment in Realism Evaluation: Professional vs. Amateur.

```
3) Appropriateness (Context-Dependent)
Tonal fit to the prompt's role & scene (stereotype-based).
If ANY portion before 30s is ill-fitting, choose the LOWER
score.
Ignore AudioQuality and generic naturalness here|judge only
contextual tone/demeanor/status/urgency/emotion.
Scale:
- 5 (Completely Appropriate):  Fits both role AND scene
perfectly; fully aligned with emotion, social dynamics,
status, and events; immersive and consistent.
- 4 (Mostly Appropriate):  Strong match with minor flaws
(e.g., slight intensity/timing/subtlety issues) that do not
break alignment.
- 3 (Adequately Appropriate):  Broad fit to role/scene but
lacks depth or shows mild inconsistencies in key elements.
- 2 (Slightly Appropriate):  Only role OR scene fit is
acceptable; superficial/partial alignment; missing/mismatched
cues.
- 1 (Completely Inappropriate):  Fails role AND scene;
tone/status/urgency clearly mismatched.
```

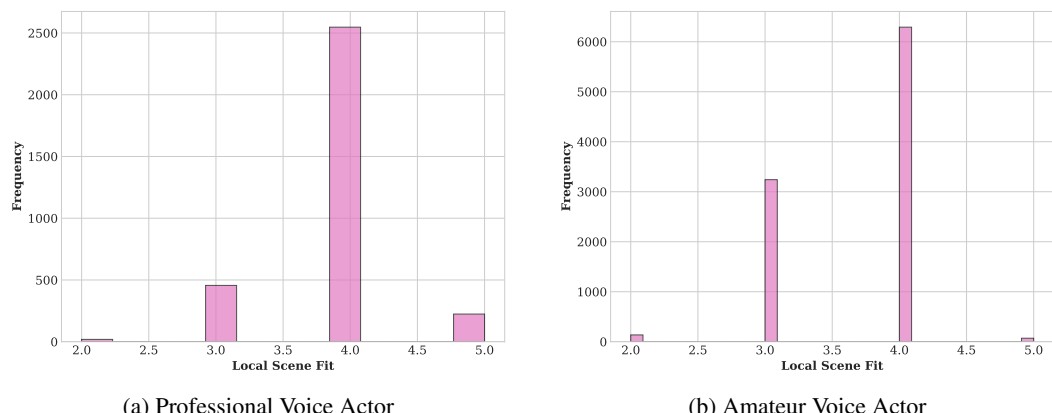

(a) Professional Voice Actor      (b) Amateur Voice Actor

Figure 16: Distribution of Local Scene Fit in Realism Evaluation: Professional vs. Amateur.

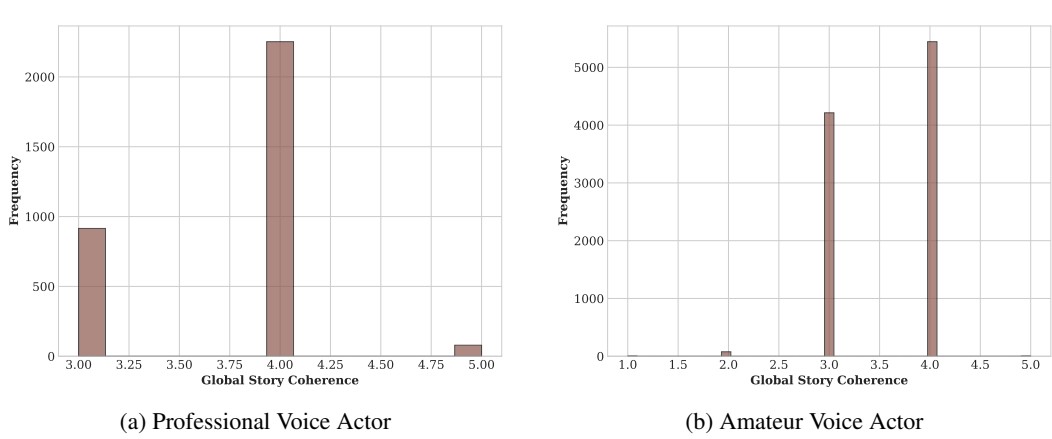

(a) Professional Voice Actor      (b) Amateur Voice Actor

Figure 17: Distribution of Global Story Coherence in Realism Evaluation: Professional vs. Amateur.

---

**OUTPUT SCHEMA (minified JSON)**
`{"content_pass":true/false,"audio_quality":1-5,"human_likeness":1-5,"appropriatenes`

---

**Rubrics Prompts (Realism Evaluation)**

**WHAT TO DO**
1. **Read first:** Global Profile ⇒ Local Scene.
2. **Play** the audio **while skimming** the transcript (transcript
may be imperfect).
3. **Focus only on the target speaker**.
   -- Ignore background SFX / music / other voices except
where they influence the target character's timing or tone.
4. **Apply the rubric below EXACTLY** and return **only** a minified
JSON object { no prose, line-breaks, or extra keys (valid for
json.loads).
5. **Scale is 1 (poor) ... 5 (excellent)**. Use integers only.

**DIMENSIONS & RULES**
For every dimension, 5 = best possible, 1 = worst. Use these
anchor descriptions to choose the closest integer.

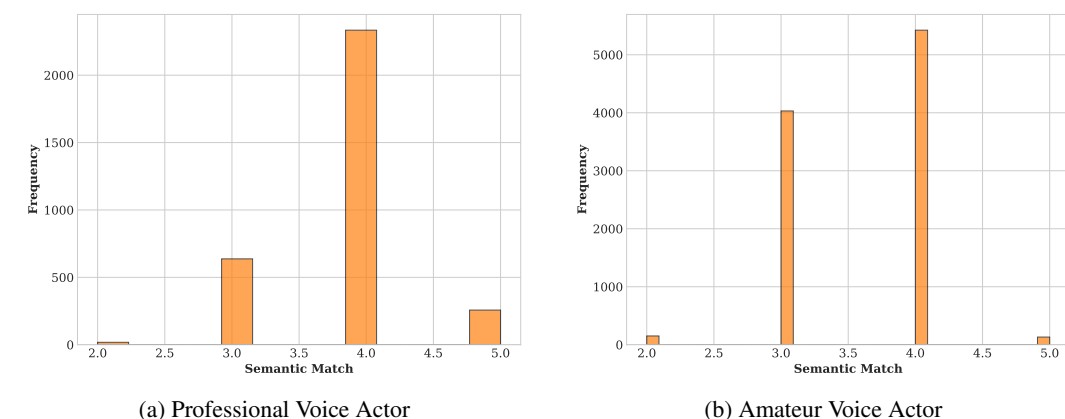

(a) Professional Voice Actor        (b) Amateur Voice Actor

Figure 18: Distribution of Semantic Match in Realism Evaluation: Professional vs. Amateur.

```
1) PitchDynamics
5:  Natural, situation-appropriate melodic movement; subtle,
believable inflections.
4:  Generally varied and fitting, minor stiffness or flat
spots.
3:  Neutral or limited inflection; neither distracting nor
expressive.
2:  Forced or sudden shifts that feel \performed."
1:  Monotone or blatantly mis-pitched for the scene.

2) RhythmicNaturalness
5:  Flowing, lifelike rhythm; natural pauses.
4:  Mostly smooth; slight hiccups.
3:  Even pacing; read-aloud style.
2:  Halting, mechanical, broken phrases.
1:  Robotic timing, destroys immersion.

3) StressEmphasis
5:  Perfect stress on key words; enhances meaning.
4:  Generally correct with few mistakes.
3:  Mostly flat; generic sentential stress.
2:  Over-emphasis or misplaced stress.
1:  Flat OR every word equally stressed.

4) EmotionAccuracy
5:  Emotion fits precisely; subtle, authentic.
4:  Appropriate but slightly under/overstated.
3:  Neutral, misses clear opportunity.
2:  Noticeably off (e.g., joking tone in crisis).
1:  Jarring or nonsensical emotion.

5) EmotionIntensity (only if 4 ≥ 3)
5:  Perfectly calibrated.
4:  Minor overshoot/undershoot.
```

```
3:  Mildly too strong/weak.
2:  Clearly over- or under-acted.
1:  Wildly inappropriate.

6) EmotionalDynamicRange (only if 5 ≥ 3)
5:  Nuanced, organic shifts.
4:  Variation present, somewhat constrained.
3:  Mostly steady; slight modulation.
2:  Flat OR abrupt jumps.
1:  Chaotic, inconsistent emotion.

7) VoiceIdentityMatching
5:  Perfectly matches profile.
4:  Acceptable, minor mismatch.
3:  Generic voice.
2:  Noticeable mismatch.
1:  Obviously wrong identity.

8) TraitEmbodiment
5:  Core traits audible, believable.
4:  Most traits present, one weak.
3:  Traits faint, neutral delivery.
2:  Traits clash or unclear.
1:  No traits or fully off.

9) LocalSceneFit
5:  Perfect alignment with objective & mood.
4:  Minor mismatch.
3:  Neutral fit.
2:  Somewhat confusing.
1:  Contradicts scene.

10) GlobalStoryFit
5:  Consistent with character arc.
4:  Slightly atypical but plausible.
3:  Generic filler.
2:  Conflicts with backstory.
1:  Breaks identity entirely.

11) SemanticMatchness
5:  Perfectly advances scene.
4:  Minor mismatch but acceptable.
3:  Neutral filler.
2:  Partially conflicts with scene.
1:  Directly contradicts/derails.

Penalty Rule:  If the speaker is clearly the wrong person,
set voice_identity_matching = 1 and trait_embodiment = 1.
```

```
OUTPUT SCHEMA (minified JSON)
{ "pitch_dynamics":1-5, "rhythmic_naturalness":1-5,
"stress_emphasis":1-5, "emotion_accuracy":1-5,
"emotion_intensity":1-5, "emotional_dynamic_range":1-5,
"voice_identity_matching":1-5, "trait_embodiment":1-5,
"local_scene_fit":1-5, "global_story_fit":1-5,
"semantic_matchness":1-5 }
```

For zeroshot and fewshot settings, we condition the model with the rubric prompt and then append few-shot demonstration triplets (input, audio, gold JSON) to mirror the human annotation protocol. As shown in Listing 2, We use `Gemini2.5Pro` for these few-shot exemplars and require a minified JSON output consistent with the rubric schema.

Listing 2: Prompt construction for zeroshot and fewshot experiments (Gemini 2.5 Pro).

```python
import json
from typing import List, Dict, Literal

Mode = Literal["archetype", "realism"]

def _minify_json(d: Dict) -> str:
    return json.dumps(d, separators=(",", ":"))

def _format_input_desc(sample: Dict, mode: Mode) -> str:
    if mode == "archetype":
        q = sample.get("question", "")
        return f"Prompt's role and scene: {q}"
    gp = sample.get("char_profile", "")
    ls = sample.get("local_scene", "")
    cs = sample.get("char_style", "")
    return f"Global Profile: {gp}; Local Scene: {ls}; Traits: {cs}"

def _audio_part_from_wav(path: str, wav_to_base64) -> Dict:
    b64 = wav_to_base64(path)
    return {
        "type": "input_audio",
        "input_audio": {"data": "data:audio/wav;base64," + b64,
            "format": "wav"},
    }

def build_prompt_with_examples(
    rubric_prompt: str,
    few_shot_examples: List[Dict],
    current_item: Dict,
    mode: Mode,
    wav_to_base64,
    model_name: str = "gemini-2.5-pro",
) -> List[Dict]:
    """Constructs rubric + few-shot + current eval into a single
        prompt."""
    parts: List[Dict] = []

    # 1) Rubric
    parts.append({"type": "text", "text": rubric_prompt.strip() +
        "\n\n"})

    # 2) Few-shot demonstrations
    for i, ex in enumerate(few_shot_examples, start=1):
        parts.append({"type": "text", "text": f"**Few-shot Example
            {i}**\n\n"})
        parts.append({"type": "text", "text": f"**Example Input:**
            {_format_input_desc(ex, mode)}\n\n"})
        parts.append(_audio_part_from_wav(ex["wav_path"], wav_to_base64))
```

```
44        if "annotation_prompt" in ex:
45            gold_json = _minify_json(ex["annotation_prompt"])
46            parts.append({"type": "text", "text": f"\n\n**Example
                  Output:** {gold_json}\n\n"})
47
48    # 3) Current evaluation
49    parts.append({"type": "text", "text": "**Current Evaluation**\n\n"})
50    parts.append({"type": "text", "text": f"**Current Input:**
          {_format_input_desc(current_item, mode)}\n\n"})
51    parts.append(_audio_part_from_wav(current_item["wav_path"],
          wav_to_base64))
52    parts.append({"type": "text",
53                  "text": "\n\nPlease evaluate the current audio and
                      return ONLY a minified JSON."})
54
55    return [
56        {"role": "system", "content": f"You are a helpful assistant
              using {model_name}."},
57        {"role": "user", "content": parts},
58    ]
```

For zeroshot and few-shot experiments on commercial models such as `Gemini` and `GPT-4o`, we report results as the average of five independent runs to reduce randomness and obtain more stable predictions. In contrast, for open-source models, we observed that their instruction-following ability is relatively weak. To mitigate this, we prompt them to predict one dimension at a time rather than all dimensions jointly. Specifically, for each metric, after showing the rubric prompt, we request completion in a fixed template and extract the probability distribution over the discrete tokens `[1]`–`[5]`. The final score is then computed from these probabilities, which we found makes open-source models usable to some extent under this setting.

For few-shot experiments, we randomly sample three exemplars from the training set, with each exemplar chosen such that its human scores are close to the rounded average across all annotators. These exemplars are appended as demonstrations in the prompt. As with the zeroshot case, we run five trials for commercial models and report the averaged results for evaluation.

### L.2 ZEROSHOT/FEWSHOT ABLATION EXPERIMENTS

Table 11 shows how different prompting strategies affect realism evaluation performance with `Gemini2.5Pro`. Several observations can be drawn:

- Effect of repeated runs. When comparing single-run decoding (`Zeroshot-1run`) against averaging over five runs (`Zeroshot`), we see consistent gains across nearly all dimensions (average score improves from 0.353 to 0.390). This suggests that sampling variance plays a substantial role in the reliability of large language model (LLM) evaluators, and averaging multiple outputs reduces this variance, producing more stable correlation with human annotations.

- Impact of few-shot conditioning. Introducing even three examples (`3-Shot`) yields a clear improvement over zeroshot (0.433 vs. 0.390). This indicates that providing explicit annotated demonstrations helps the model better align with the rubric. Interestingly, when using only a single run with few-shot demonstrations (`3-Shot-1run`), the performance is only marginally better than pure zeroshot, highlighting that both averaging and demonstrations are complementary: demonstrations improve grounding, while multiple runs mitigate stochasticity.

- Scaling the number of demonstrations. Increasing the few-shot set size from three to six and nine further boosts performance (0.459 and 0.469, respectively). The improvements are most pronounced in prosodic and emotional categories (e.g., *Pitch*, *Accuracy*, *Intensity*), suggesting that more examples provide stronger guidance in modeling subtle paralinguistic cues. Gains in *Character Consistency* and *Contextual Relevance* are also noticeable, although smaller in magnitude, which may indicate that these higher-level judgments are less sensitive to prompt scaling once a few demonstrations are provided.

Table 11: Realism role-play evaluator performance with `Gemini2.5Pro` in different fewshot conditions. Here, "1run" stands for only conduct generation for once.

| Config. | Prosodic Dynamics | | | Emotional Expressiveness | | | Character Consistency | | Contextual Relevance | | Avg. |
|---|---|---|---|---|---|---|---|---|---|---|---|
| | Pitch | Rynthm | Stress | Accuracy | Intensity | Transition | Identity | Traits | Local | Global | |
| Zeroshot-1run | 0.277 | 0.325 | 0.303 | 0.355 | 0.286 | 0.295 | 0.480 | 0.475 | 0.308 | 0.424 | 0.353 |
| Zeroshot | 0.319 | 0.352 | 0.336 | 0.391 | 0.316 | 0.336 | 0.529 | 0.509 | 0.349 | 0.465 | 0.390 |
| 3-Shot-1run | 0.316 | 0.335 | 0.323 | 0.358 | 0.279 | 0.294 | 0.530 | 0.476 | 0.335 | 0.428 | 0.366 |
| 3-Shot | 0.400 | 0.450 | 0.432 | 0.414 | 0.344 | 0.353 | 0.584 | 0.505 | 0.374 | 0.477 | 0.433 |
| 6-Shot | 0.419 | 0.431 | 0.427 | 0.455 | 0.393 | 0.391 | 0.606 | 0.540 | 0.404 | 0.519 | 0.459 |
| 9-Shot | **0.419** | **0.453** | **0.434** | **0.464** | **0.404** | **0.416** | **0.613** | **0.551** | **0.407** | **0.524** | **0.469** |

In general, the results demonstrate two complementary strategies for stabilizing LLM-based evaluators: (i) averaging multiple runs to reduce randomness, and (ii) enriching prompts with more few-shot exemplars to improve adherence to rubric-based evaluation. Combining both leads to the strongest overall performance, with the nine-shot condition achieving the highest average correlation (0.469).

### L.3 FINE-TUNING SETUP FOR EVALUATION MODELS

For supervised fine-tuning, we primarily adopted a parameter-efficient strategy using LoRA, applied to the `Qwen2Audio-7B-Instruct` model (Chu et al., 2024). LoRA was configured with a rank of 16, $\alpha = 32$, and a dropout of 0.1, targeting the major projection layers, and trained for 10,000 steps to ensure stable convergence. The training pipeline employed a per-device batch size of 4 with gradient accumulation of 4 steps (effective batch size 32), AdamW optimization with cosine scheduling, a peak learning rate of $5 \times 10^{-5}$, weight decay of 0.01, and a warmup strategy defined by 500 steps. Gradient clipping with a norm of 1.0, `bf16` precision, and gradient checkpointing were enabled to stabilize training and reduce memory overhead. Distributed training was performed on 2xH100 GPUs using DeepSpeed (ZeRO stage-1).

While we utilize various prompts and sample them randomly during the fine-tuning experiments, we provide an example prompt as follows:

---

**Example Prompts in Realism Evaluation (Pitch Dynamics Dimension)**

```
You are an expert evaluator of speech delivery and
storytelling.

Based on the given information, rate the following SINGLE
dimension on a 1{5 scale (1 = poor, 5 = excellent).
Return ONLY a number between 1 and 5.

Global Profile: {global_profile}
Local Scene: {local_story}
Available Traits: {available_traits}

Dimension: PitchDynamics | Variety and appropriateness of
pitch contours.

5:  Natural, situation-appropriate melodic movement; subtle,
believable inflections that reveal emotion or emphasis.
4:  Generally varied and fitting, but with minor stiffness or
occasional flat spots.
3:  Neutral or limited inflection; neither distracting nor
especially expressive.
2:  Forced, exaggerated, or oddly sudden shifts that feel
'performed' rather than lived.
1:  Monotone or blatantly mis-pitched for the scene (e.g.,
cheerfully high in a funeral).
```

---

Table 12: Archetype role-play evaluator performance. We report classification accuracy for *Content Pass* and Pearson correlation coefficients for other dimensions. The Avg. column is the average of all correlation coefficients.

| Config. | Content Pass Acc. | Audio Quality | Human Likeness | Appropriateness | Avg. |
|---|---|---|---|---|---|
| `Base-FT` | 92.1% | **0.688** | 0.648 | **0.551** | 0.616 |
| `Base-LoRA` | 92.6% | 0.668 | 0.648 | 0.545 | 0.620 |
| `Instruct-FT` | 92.6% | 0.650 | 0.684 | 0.492 | 0.609 |
| `Instruct-LoRA` | **93.6%** | 0.682 | **0.680** | 0.525 | **0.629** |

Table 13: Realism role-play evaluator performance (basic test set). We report Pearson correlation coefficients for other dimensions. The Avg. column is the average of all coefficients.

| Config. | Prosodic Dynamics | | | Emotional Expressiveness | | | Character Consistency | | Contextual Relevance | | Avg. |
|---|---|---|---|---|---|---|---|---|---|---|---|
| | Pitch | Rynthm | Stress | Accuracy | Intensity | Transition | Identity | Traits | Local | Global | |
| `Base-FT` | 0.643 | 0.633 | 0.655 | 0.657 | 0.600 | 0.599 | 0.689 | 0.668 | 0.611 | 0.707 | 0.646 |
| `Base-LoRA` | 0.628 | 0.633 | 0.643 | 0.637 | 0.586 | 0.590 | 0.668 | 0.651 | 0.578 | 0.684 | 0.630 |
| `Instruct-FT` | **0.657** | **0.655** | **0.675** | **0.674** | **0.627** | **0.626** | **0.752** | **0.685** | **0.615** | **0.714** | **0.668** |
| `Instruct-LoRA` | 0.621 | 0.627 | 0.631 | 0.632 | 0.596 | 0.601 | 0.660 | 0.655 | 0.563 | 0.668 | 0.625 |

```
Example output:  4"

Output the answer in <answer> </answer>.
```

## L.4  FINE-TUNING ABLATION EXPERIMENTS

Tables 12, 13, and 14 present ablation studies across different fine-tuning strategies, including the use of `Qwen2Audio-7B-Base` or `Qwen2Audio-7B-Instruct`; and the use of fully fine-tuning or LoRA fine-tuning. We use `Base-FT`, `Base-LoRA`, `Instruct-FT`, and `Instruct-LoRA` to represents different configuration settings in following experiments.

**Archetype evaluation.** On the archetype test set, all fine-tuned variants achieve strong *Content Pass* accuracy above 92%. Differences appear mainly in correlation-based metrics. Here, `Base-FT` yields the highest score on *Audio Quality* and *Appropriateness*, while `Instruct-LoRA` delivers the strongest performance in *Human Likeness* and the best overall average (0.629). This suggests that instruction-tuned checkpoints paired with parameter-efficient adaptation can better capture higher-level perceptual traits.

**Realism evaluation (basic set).** For realism evaluation on the basic test set, `Instruct-FT` consistently achieves the strongest correlations across most dimensions, especially for *Character Identity* and *Traits*, leading to the best average correlation (0.668). `Base-LoRA` also performs competitively (0.630), while `Instruct-LoRA` lags slightly behind (0.625). These results highlight that full fine-tuning may still offer an advantage when the evaluation set is relatively homogeneous and closer to the training distribution.

**Realism evaluation (real recordings).** A different pattern emerges on the real recording test set. Both `Base-LoRA` and `Instruct-FT` struggle, yielding negative or near-zero correlations across most dimensions. In contrast, `Instruct-LoRA` shows clear improvements, achieving positive correlations in all prosodic and emotional dimensions and producing the highest overall average (0.247). This indicates that LoRA adaptation on top of an instruction-tuned backbone provides better generalization to out-of-distribution, real-world data.

**Overall summary.** While `Instruct-FT` is competitive on in-domain evaluation, it suffers a substantial drop when applied to real recordings. By contrast, `Instruct-LoRA` offers a more robust balance: it maintains reasonable performance on archetype and basic sets, while significantly outperforming other settings on real-world data. Based on this robustness, we adopt `Instruct-LoRA` as our final configuration for subsequent experiments.

Table 14: Realism role-play evaluator performance (real recording test set). We report Pearson correlation coefficients for other dimensions. The Avg. column is the average of all coefficients.

| Config. | Prosodic Dynamics | | | Emotional Expressiveness | | | Character Consistency | | Contextual Relevance | | Avg. |
|---|---|---|---|---|---|---|---|---|---|---|---|
| | Pitch | Rynthm | Stress | Accuracy | Intensity | Transition | Identity | Traits | Local | Global | |
| Base-FT | -0.422 | -0.119 | -0.333 | -0.261 | -0.330 | -0.351 | 0.059 | 0.538 | -0.3083 | -0.153 | -0.168 |
| Base-LoRA | -0.372 | -0.016 | -0.304 | -0.195 | -0.273 | -0.249 | -0.091 | 0.458 | -0.291 | -0.146 | -0.148 |
| Instruct-FT | -0.269 | -0.031 | -0.202 | -0.208 | -0.222 | -0.242 | **0.076** | **0.543** | -0.219 | -0.094 | -0.087 |
| Instruct-LoRA | **0.288** | **0.270** | **0.362** | **0.277** | **0.331** | **0.397** | -0.034 | 0.077 | 0.217 | **0.279** | **0.247** |

# M    LIMITATIONS AND FUTURE WORKS

Despite the comprehensive design of Speech-DRAME, several limitations remain. First, while our dual evaluation paradigm combines top-down archetype assessment with bottom-up realism evaluation, both strategies are constrained by the quality and scope of available data. Archetype evaluation inevitably relies on simplified stereotypes, which can overlook subtler paralinguistic and contextual cues. Realism evaluation, though richer, still struggles with domain mismatch between synthetic data and real human recordings, as evidenced by the performance drop observed in the real-recording test set. This highlights the persistent gap between benchmark-driven evaluation and real-world variability in speech role-play.

Second, the present framework emphasizes bilingual evaluation (Mandarin and English), but linguistic and cultural diversity is far broader. Cross-lingual generalization and culturally sensitive evaluation are still open challenges, particularly since we observed annotation discrepancies across language groups. This limits the extent to which Speech-DRAME can claim universal coverage.

Third, while DRAME-Eval substantially outperforms zero-shot and few-shot ALLMs, its reliance on fine-tuning from a strong base model raises questions of accessibility and scalability. Future work should explore lightweight adaptation techniques, semi-supervised training with partially annotated data, and leveraging emergent capabilities of open-source multimodal LLMs.

Finally, although we validated DRAME-Eval against human annotations, alignment under realism settings remains modest. This suggests that even human-annotated benchmarks are insufficiently fine-grained to capture the full complexity of human perception in speech role-play. Richer annotation protocols, more diverse speaker populations, and integration of perceptual psychology insights could further improve evaluation fidelity.

Looking forward, we see several promising research directions. We intentionally restrict evaluation to single-turn responses for clarity and tractability. However, multi-turn extensions that capture narrative progression, long-term consistency, and dynamic emotion shifts remain a natural next step. Incorporating multilingual and cross-cultural datasets will enhance generality and fairness. On the modeling side, exploring hybrid evaluation strategies that combine discrete rubrics with continuous perceptual signals (e.g., prosody embeddings or affective measures) may better bridge the gap between automatic and human judgments. Finally, integrating Speech-DRAME into reinforcement learning pipelines as a reward model could help drive the next generation of expressive, human-aligned speech role-play systems.

