# OpenReview forum: "Speech-DRAME: A Framework for Human-Aligned Benchmarks in Speech Role-Play"
_ICLR.cc/2026/Conference — ICLR 2026 Conference Withdrawn Submission_

### Official Review · Reviewer_xKtL · 2025-10-31

**Soundness:** 1
**Presentation:** 1
**Contribution:** 1
**Rating:** 0
**Confidence:** 3

**Summary:**

Exceeded the page limit. Desk Reject.

**Strengths:**

Exceeded the page limit. Desk Reject.

**Weaknesses:**

Exceeded the page limit. Desk Reject.

**Questions:**

Exceeded the page limit. Desk Reject.

---

### Official Review · Reviewer_MmWZ · 2025-11-01

**Soundness:** 1
**Presentation:** 1
**Contribution:** 1
**Rating:** 0
**Confidence:** 5

**Summary:**

The paper is desk-rejected since at the time of submission, the main text exceeds 9 pages.

**Strengths:**

The paper is desk-rejected since at the time of submission, the main text exceeds 9 pages.

**Weaknesses:**

The paper is desk-rejected since at the time of submission, the main text exceeds 9 pages.

**Questions:**

The paper is desk-rejected since at the time of submission, the main text exceeds 9 pages.

---

### Official Review · Reviewer_QBf4 · 2025-11-01

**Soundness:** 3
**Presentation:** 2
**Contribution:** 2
**Rating:** 4
**Confidence:** 4

**Summary:**

The paper introduces Speech-Dream, a framework for benchmarking speech role-play, which includes an evaluation benchmark for training and testing speech evaluation models, a fine-tuned evaluation model (DREAM-Eval) that outperforms zero-shot and few-shot Audio LLMs, and a speech role-play benchmark for comparing speech foundation models. Speech-Dream proposes two complementary evaluation strategies: Archetype Evaluation (based on synthetic data) and Realism Evaluation (based on real speech data). Compared to Audio LLMs, DREAM-Eval achieves stronger alignment with human ratings.

**Strengths:**

1. The paper proposes two complementary evaluation strategies: Archetype Evaluation (synthetic data-based) and Realism Evaluation (real human speech-based).
2. It builds a comprehensive framework, including datasets, evaluation models, and benchmarks, and evaluates multiple proprietary and open-source models.
3. The fine-tuned Qwen2Audio model achieves superior evaluation quality compared to general-purpose Audio LLMs.

**Weaknesses:**

1. Realism-based evaluation suffers from domain mismatch, as its training data also contains synthetic speech, reducing its usability despite being based on real human speech.
2. Realism Evaluation shows poor alignment with human perception (Spearman correlation of only 0.375), indicating limited reliability.
3. The framework only supports single-turn evaluations, failing to capture the coherence of multi-turn narratives.
4. The paper lacks audible demo cases for the speech role-play generation task.
5. Some parts of the paper are less concise, with redundant content, such as lines 083 to 101 in the introduction.

**Questions:**

Please address the issues described in the Weaknesses section. Resolving these concerns could improve the paper’s evaluation.

---

### Official Review · Reviewer_f2WH · 2025-11-01

**Soundness:** 2
**Presentation:** 1
**Contribution:** 2
**Rating:** 4
**Confidence:** 4

**Summary:**

This paper proposes Speech-DRAME, a framework for evaluating speech role-play tasks using both generative models and evaluation models (SEMs). The authors aim to build a comprehensive benchmark, consisting of the EvalBench and RoleBench, for evaluating role-play performance in speech generation. They also introduce a dual evaluation strategy (Archetype-based and Realism-based evaluations) and argue that their method provides an improvement over previous evaluation techniques based on zero-shot large language models (ALLMs).

**Strengths:**

1. **Novel Framework**: The proposed framework offers an interesting approach by integrating role-play generation with a dual evaluation strategy. The introduction of EvalBench and RoleBench datasets provides a clear framework for the evaluation of speech-based role-playing tasks.
2. **Detailed Benchmark Design**: The inclusion of both Archetype and Realism evaluation strategies ensures that the proposed method addresses both large-scale and fine-grained human perception of speech quality.
3. **Relevance to the Community**: The focus on improving role-play evaluation and the framework's potential to influence future work in speech generation and assessment makes it relevant to the community.

**Weaknesses:**

1. **Over-reliance on the Appendix**: A significant amount of important information is placed in the **Appendix**, which makes it difficult to follow the main arguments and understand the contributions in the body of the paper. A well-written paper should be **self-contained**, with all critical information included in the main text.
2. **Clarity of Motivation**: The paper lacks a clear motivation regarding the limitation of **zero-shot ALLMs** as evaluation judges. There is insufficient discussion of the limitations of using ALLMs for this task.
   - In line 45-46, the statement **"this strategy"** is vague and lacks clarity regarding what exactly is being referred to.
   - The statement in **Lines 131-132** ("Yet studies consistently reveal high prompt sensitivity and shallow capture of paralinguistic cues") is **unsupported** and should be backed by references or experimental evidence.
3. **Clarity of Realism-Based Evaluation**:
   - The **Realism-based role-play evaluation** section is not explained clearly. The paper mentions it but does not provide enough details about how it differs from the Archetype evaluation or what specific criteria it evaluates.
   - The paper also mentions the **dual evaluation strategy** but does not sufficiently explain the distinctions and advantages of each evaluation approach (Archetype vs. Realism).
4. **Fine-tuning Data Quality and Diversity**:
   - While the paper mentions the use of **fine-tuning**, it does not address the **quality control** or **diversity** of the fine-tuning data. Given that the fine-tuned model is trained on data from the same distribution, the results may be overly optimistic and not fully generalizable.
   - In **Line 240**, the paper mentions that data is collected from **speech foundation models in D.4**, but some of the models (e.g., Qwen-2.5-Omni) are being used for evaluation, which could affect the fairness of the evaluation.


Several other weaknesses:

1. **Use of TTS-Generated Data for Archetype**
   - The decision to use **TTS-generated role-play data** instead of **real human speech** in the **Archetype-based evaluation** is not justified sufficiently. Given that real human speech data is available (e.g., from firefighter media sources) and the main claim line053, it is unclear why synthetic data was chosen.

2. **Evaluation Across Unified Prompting Template**:

   - The paper claims that **all evaluations follow a unified prompting template**, but this approach may not fully reflect the model's ability to handle diverse inputs and scenarios. This could limit the generality of the evaluation.

   - **Suggested improvement**: The authors should discuss the potential limitations of using a single prompting template and consider testing the model across varied inputs to better assess its performance.

**Questions:**

The paper presents an interesting framework with potential to advance the field of speech role-play evaluation. However, the paper could benefit from clearer explanations and justifications for certain methodological choices, particularly regarding the use of TTS-generated data, the zero-shot ALLM evaluation model, and the definition of key terms. Additionally, a stronger comparison with prior work and more detailed experimental design would strengthen the paper.

Additionally, the **experimental design** could be further detailed, particularly with regard to **fine-tuning data** and its potential impact on generalizability.

From my perspective, the **training of evaluators** (e.g., the DRAME-Eval model) is one of the most valuable aspects of this paper, but there is a need for further exploration of its **generalization across different domains** and **evaluation metrics**.

---

### Note · Authors · 2025-11-13

**Comment:**

Dear Area Chairs and Reviewers,

Thank you for evaluating our submission. Due to a formatting oversight on our part, the paper exceeded the page limit by two lines, and we fully understand the resulting desk-reject recommendations. We would therefore like to formally withdraw the paper.

We would especially like to thank the two reviewers who provided detailed and thoughtful feedback despite the formatting issue. Their comments were highly constructive, clearly engaged with the technical substance of our work, and have already helped us identify several important directions for improvement. We are truly grateful for the time and care they invested in offering such comprehensive critiques.

Thank you again for your understanding and for the reviewers’ valuable efforts.

Sincerely,
Speech-DRAME Authors

**Withdrawal Confirmation:**

I have read and agree with the venue's withdrawal policy on behalf of myself and my co-authors.